# Anti-Tumor Effect of Turandot Proteins Induced via the JAK/STAT Pathway in the *mxc* Hematopoietic Tumor Mutant in *Drosophila*

**DOI:** 10.3390/cells12162047

**Published:** 2023-08-11

**Authors:** Yuriko Kinoshita, Naoka Shiratsuchi, Mayo Araki, Yoshihiro H. Inoue

**Affiliations:** Biomedical Research Center, Kyoto Institute of Technology, Mastugasaki, Kyoto 606-0962, Japan; m2641013@edu.kit.jp (Y.K.); shiratsuchi.naoka.22n@st.kyoto-u.ac.jp (N.S.); mayo.no.17@gmail.com (M.A.)

**Keywords:** JAK/STAT, Turandots, mxc, drosophila, apoptosis, antitumor

## Abstract

Several antimicrobial peptides suppress the growth of lymph gland (LG) tumors in *Drosophila multi sex comb* (*mxc*) mutant larvae. The activity of another family of polypeptides, called Turandots, is also induced via the JAK/STAT pathway after bacterial infection; however, their influence on *Drosophila* tumors remains unclear. The JAK/STAT pathway was activated in LG tumors, fat body, and circulating hemocytes of mutant larvae. The mRNA levels of Turandot (*Tot*) genes increased markedly in the mutant fat body and declined upon silencing Stat92E in the fat body, indicating the involvement of the JAK/STAT pathway. Furthermore, significantly enhanced tumor growth upon a fat-body-specific silencing of the mRNAs demonstrated the antitumor effects of these proteins. The proteins were found to be incorporated into small vesicles in mutant circulating hemocytes (as previously reported for several antimicrobial peptides) but not normal cells. In addition, more hemocytes containing these proteins were found to be associated with tumors. The mutant LGs contained activated effector caspases, and a fat-body-specific silencing of Tots inhibited apoptosis and increased the number of mitotic cells in the LG, thereby suggesting that the proteins inhibited tumor cell proliferation. Thus, Tot proteins possibly exhibit antitumor effects via the induction of apoptosis and inhibition of cell proliferation.

## 1. Introduction

Invertebrates such as *Drosophila* do not possess an acquired immune system but use only the innate immune system for their biological defense. The *Drosophila* immune system comprises a cellular and a humoral immunity [1,2]. Cellular immunity is triggered by means of phagocytosis and hemocyte melanization, when foreign substances invade the body. Humoral immunity involves the production of antimicrobial peptides (AMPeps) in the fat body to eliminate foreign substances [3,4,5]. More than seven primary AMPeps are present in *Drosophila*. When fungi or Gram-positive bacteria invade the *Drosophila* body, the Toll pathway is activated, resulting in the induction of five AMPeps: domycin, metchnikowin, defensin, attacin, and cecropin. In contrast, when Gram-negative bacteria invade, the Imd pathway is activated, resulting in the induction of four typical AMPeps: attacin, cecropin, diptericin, and drosocin [6]. Recent studies have also suggested that these AMPeps attack foreign substances and tumor cells originating from normal cells [7,8]. In addition, the Janus kinase (JAK)/signal transducer and activator of transcription (STAT) signaling pathway is activated in response to foreign microorganisms and cellular stress [9,10]. 

Upd3, which is a functional ortholog of mammalian interleukin-6, is induced in response to bacterial and viral infections in *Drosophila* [11,12]. The cytokine is explicitly expressed in circulating hemocytes in response to invasion by foreign organisms [13]. Upd3 binds to its receptor, Domeless, on the fat body and dimerizes with the receptor, resulting in the phosphorylation of JAK encoded by *hop* bound to its intracellular domain. Furthermore, Stat92E binds to the phosphorylation domain of Hop and is phosphorylated. The activated Stat92E enters the nucleus, subsequently inducing the expression of several target genes, including some members of the *Turandot* (*Tot*) family [14,15,16,17]. One of the *Tot* family genes, *TotA*, is known to be induced by bacterial infection and other stresses, via the JAK/STAT signaling pathway [13,16,18]. However, few studies have yet investigated whether the *Tot* genes were induced in response to *Drosophila* tumors or exerted antitumor effects.

A lethal mutant of the *multi sex comb* (*mxc*) gene, *mxc^mbn1^*, has been recognized to cause hematopoietic tumor in *Drosophila* [7,19,20,21,22,23]. *mxc^mbn1^* mutants exhibit hyperplasia in the larval hematopoietic tissue called the lymph gland (LG). Immature cells expressing Upd3 in the medullary zone of the LG overproliferate in the hyperplastic tissues of mutant larvae [7]. Mutant larvae also contain an increased number of circulating hemocytes and abnormally differentiated hemocytes in the hemolymph. Furthermore, LG cells isolated from mutant larvae could further proliferate in the host abdomen when implanted into normal adults [19,20] (Takarada, Kinoshita, and Inoue, submitted), while wild-type LG cells did not proliferate in the host. Thus, LG tumors in *mxc^mbn1^* mutants can be considered malignant [7,19,20,21]. Furthermore, excessive circulating hemocytes and abnormally differentiated hemocytes have been observed in the mutant hemolymph [19,20,22]. Based on this evidence, we concluded that *mxc^mbn1^* larvae showed leukemia-like phenotypes, as observed in mammals [21,22].

In this study, we demonstrated the induction of Tot proteins via the JAK/STAT pathway and the antitumor effect of the proteins on LG tumors in *Drosophila mxc* mutants. We then performed genetic analyses to elucidate the mechanisms underlying the induction and antitumor effects of the proteins. First, we confirmed that many abnormal hemocytes that continued to express the *upd3* marker were present in the mutant hemolymph. Next, we investigated whether the JAK/STAT pathway downstream of the ligand was activated in the mutant larvae. We further investigated whether the activation of the pathway induced the transcription of its target genes encoding Tot proteins in the fat body of *mxc^mbn1^* larvae. Subsequently, we examined whether Tot proteins exerted antitumor effects by suppressing tumor growth and inducing apoptosis in the tumor. Finally, we elucidated the involvement of hemocytes in the production of the Tot protein in the fat body of mutant larvae harboring LG tumors. Based on these genetic data, we propose a model for the mechanism underlying the induction of the proteins from the fat body and their antitumor effect. Our findings will help understand the function of the innate immune system in response to tumors and could contribute to the development of new antitumor medicines.

## 2. Materials and Methods

### 2.1. Drosophila Stocks and Husbandry

All stocks were maintained on standard cornmeal food as described previously [24]. *w^1118^* (abbreviated as *w*) was used as a normal control stock. A recessive lethal allele of *mxc* showing the LG tumor phenotype, *mxc^mbn1^*, was obtained from the Bloomington Drosophila Stock Center (BDSC, Bloomington, IN, USA) [7,21]. The following Gal4 driver stocks were used for ectopic expression in specific larval tissues: *P{w[+mC]=r4-GAL4}3* (*r4-Gal4*) for fat-body-specific expression [25] (#BL33832; BDSC), *P{Hml-GAL4Δ}* (*Hml-Gal4*) (#BL30139; BDSC) [21] and *P{upd3-GAL4}* (*upd3-Gal4*) [26] (from N. Perrimon (Harvard Medical School, Boston, MA, USA)), for the induction of gene expression in mature hemocytes in lymph gland (LG) and circulating hemocytes, and immature hemocyte precursors in the LG, respectively. To monitor the activation of the JAK/STAT pathway, the following GFP reporter for STAT target genes was used: *PBac{y+mDint2 w^+mC=^Stat92E-GFP.FLAG}* (*Stat92E-GFP*) (#BL38670, BDSC) [27]. For dsRNA-dependent gene silencing, the following UAS-RNAi stocks were used; *P{GD6210}v14416* [28] and *P{KK112386}v106548* for *TotA* silencing (Vienna Drosophila Resource Center (VDRC), Vienna, Austria), *P{GD3091}v51123* and *P{GD3091}v51124* for *TotB* silencing (VDRC) [29], *P{KK110624}v14420* for *TotC* silencing [30], *P{GD3660}v8602* for *TotF* silencing (VDRC #v8602), and *P{KK101199}* (VDRC # *v107119*) [31] and *P{y^+t7.7^**v*
*^+t1.8^=TRiP.JF01265}attP2* for *Stat92E* silencing (BDSC#31317), while *P{w^+mC^=UAS-GFP.dsRNA.R}142* (BDSC, BL#9330) was used as a control for the RNAi experiments. To visualize the cellular localization of the TotB and TotF using anti-HA immunostaining, *M{UAS-TotB.ORF.3xHA.GW}ZH-86Fb* (FlyORF, Zurich, Switzerland (Stock Number:002780)) and *M{UAS-TotF.ORF.3xHA.GW}ZH-86Fb* (FlyORF (Stock Number:00351) were used, respectively (Bischof and FlyORF project members, 2014).

### 2.2. LG Preparation

Normal control males (*w/Y*) pupated at 6 days (28 °C) and 7 days (25 °C) after egg laying (AEL), whereas the *mxc^mbn1^* males (*mxc^mbn1^/Y*) remained in the 3rd-instar larval stage at 8 days (28 °C) and 10 days (25 °C) AEL. To minimize the possibility of a delay that may lead to hyperplastic tissue to grow, a comparative analysis of the controls and *mxc^mbn1^* mutants was performed on the same day when the wandering larvae at the 3rd instar stage was observed. Alternatively, tissues were prepared from the mutant larvae a day after lymph gland (LG) collection from the control. To avoid crowded culture condition, parent flies were transferred into a new culture vial and left there to lay eggs for 24 h. To compare the LG size, a pair of anterior lobes of the LG were collected from mature larvae and fixed in 4% paraformaldehyde for 15 min. The fixed samples were flattened with mild pressure so that the tissue was of constant thickness [7]. The size of the LG samples stained with DAPI was quantified using ImageJ version 1.47(https://imagej.nih.gov/ij/ (accessed on 24 December 2021).

### 2.3. Quantitative Real-Time PCR (qRT-PCR) Analysis

Using TRIzol reagent (Invitrogen, Waltham, MA, USA), total RNA was extracted from the 3rd-instar larvae of each genotype. cDNA was synthesized from total RNA using an oligo dT primer and a PrimeScript^TM^ High Fidelity RT-PCR Kit (TaKaRa, Clontech Laboratories, Shiga, Japan). Real-time PCR was carried out using FastStart Essential DNA Green Master Mix and a Light Cycler Nano Instrument (both from Roche, Mannheim, Germany). The qPCR primers were synthesized as follows: RP49-Fw, 5′-TTCCTGGTGCACAACGTG-3′ and RP49-Rv, 5′-TCTCCTTGCGCTTCTTGG-3′; TotA-Fw, 5′-CCCAGTTTGACCCCTGAG-3′ and TotA-Rv, 5′-GCCCTTCACACCTGGAGA-3′; TotB-Fw, 5′-CGCATGGCTCCTAGCTTAAGA-3′ and TotB-Rv, 5′-CTGGGTACTCCATCGACCATG-3′; TotC-Fw, 5′-TACTATGCCTTGCCCTGCTCC-3′ and TotC-Rv, 5′-TGTTCAGGGGACAACGTGGG-3′; TotF-Fw, 5′-AGGCACGTCAAATGCTCGC-3′ and TotF-Rv, 5′-TGTTGGTTGTTGTGTGCCCG-3′. Each sample was triplicated, and the final PCR results were obtained after averaging three biological replicates. The ∆∆Ct method was used to determine the differences between the expression of target genes relative to that of the reference gene, *Rp49*. Real-time PCR was performed on a LightCyclerNano using the FastStart Essential DNA Green master kit (Roche)

### 2.4. Immunostaining of the LGs

Larvae harboring male genital discs were selected among mature third-instar larvae. The LGs were dissected from the larvae and fixed in 4% paraformaldehyde in phosphate-buffered saline (PBS) for 15 min at room temperature. After repeated washing, the samples were blocked in blocking buffer (PBS containing 0.1% Triton X-100 and 10% normal goat serum) and the fixed samples were further incubated with primary antibodies at 4 °C overnight. Anti-HA antibody (1:200; #2367, Cell Signaling Technology, Danvers, MA, USA). After extensive washing, the specimens were incubated with Alexa 594- or Alexa 488-conjugated secondary antibodies (1:400; Molecular Probes, Eugene, OR, USA). The LG specimens were observed under a fluorescence microscope (Olympus, Tokyo, Japan, model IX81) equipped with excitation and emission filter wheels (Olympus). The specimens were illuminated with UV-filtered and shuttered light using appropriate filter wheel combinations through multiple band-passed filters. The fluorescence images were captured by a CCD camera (orcaR2, Hamamatsu Photonics, Shizuoka, Japan). Image acquisition was controlled by Metamorph software (version 7.6, Molecular Devices Japan, Tokyo, Japan) and processed using ImageJ or Photoshop CS (Adobe, San Jose, CA, USA).

### 2.5. Preparation and Immunostaining of Hemocytes

After repeated washing in PBS, a larva at the 3rd instar stage were transferred into a *Drosophila* Ringer (DR) solution (10 mM, Tris-HCl, pH 7.2, 3 mM CaCl_2_ 2H_2_O, 182 mM KCL, 46 mM NaCl) on a glass slide. The epidermis of the larvae was then cut using a set of fine forceps to allow the hemocytes in the hemolymph to be released into the DR outside the larvae. After an aliquot of DR solution containing circulating hemocytes was placed on the slide glass for evaporation, hemocytes were fixed using 4.0% paraformaldehyde for 5 min at 25 °C. Hemocyte immunostaining was performed as described above. The following primary antibodies were used: anti-P1 monoclonal antibody [31] (a gift from I. Ando, Biological Research Centre, Szeged, Hungary, 1:100), anti-L1/Attila monoclonal antibody [32] (a gift from I. Ando, 1:100), and anti-HA antibody (#2367, Cell Signaling Technology, 1:200). The specimens were then incubated with Alexa 594- or Alexa 488-conjugated secondary antibodies (1:400; Molecular Probes). Fluorescence images were acquired as described previously. They were processed with ImageJ or Adobe Photoshop CS and used to quantify the fluorescence intensity. The surface and interior of the hemocytes harboring TotB-HA were observed under Fv10i (Olympus) by altering the focus planes along the Z-axis. Subsequently, the confocal images obtained were processed using FV10-ASW Viewer software (version 4.2b) (Olympus) and Adobe photoshop CS6. Multiple images were assembled into single images using Photoshop CS6.

### 2.6. Apoptosis Assay

Apoptotic cells were detected by means of immunostaining with antiactivated effector caspase antibody. The LGs collected from the larvae were fixed in 4% paraformaldehyde for 15 min at room temperature and then repeatedly washed. The LG samples were permeabilized and blocked in blocking buffer (PBS containing 0.1% Triton X-100 and 10% normal goat serum). After several washes, LGs were immunostained with an anti-cleaved Dcp-1 antibody (1:200; #9578, Cell Signaling Technology).

### 2.7. Statistical Analysis

For the measurement of the LG area, more than 20 larvae were used for each genotype. The results are presented as bar graphs or scatter plots created using Prism (Version 9, GraphPad Software, CA, USA). The area in the pixels was calculated, and the average was determined for each LG. Each dataset was assessed using Welch’s *t*-test, an analysis of variance (ANOVA), or Fisher’s exact test. Data were tested for normality by using the Shapiro–Wilk test and normalized by the Box–Cox common transforming method. When the Box–Cox transformation could not be applied, we used the Yeo–Johnson transformation. Subsequently, Welch’s *t*-test or an ANOVA was performed using the transformed values. We used Welch’s *t*-test to compare the two groups. A one-way ANOVA followed by Bonferroni’s multiple comparisons test was applied to analyze the differences in more than two groups. A two-way ANOVA followed by Tukey’s multiple comparisons test was performed to compare the mean differences between groups split into two independent variables. Statistical significance is described in each figure legend as follows: * *p* < 0.05, ** *p* < 0.01, *** *p* < 0.001, and **** *p* < 0.0001. A *p*-value of 0.05 or less was considered statistically significant.

## 3. Results

### 3.1. The Hemolymph of the Mutant Larvae Contained Some Hemocytes with Ectopic Expression of Upd3

Previous studies have reported that undifferentiated cells expressing *upd3* overproliferate in the hyperplastic LGs of *mxc^mbn1^* larvae [7,21,22]. Therefore, we investigated whether the hemocytes in the mutant hemolymph also expressed an undifferentiated cell marker. We collected hemolymph from normal controls and *mxc^mbn1^* larvae and observed the circulating hemolymph in these samples. In the normal control, all hemocytes exhibited a nearly circular shape (Figure 1a’,c’). In contrast, hemocytes of various shapes other than the circular shape were occasionally observed in the mutant hemolymph (Figure 1b’,d’). To confirm that genetic features of circulating hemocytes changed in the mutant hemocytes, we first performed an immunostaining of circulating hemocytes in normal (*w/Y*) and mutant (*mxc^mbn1^/Y*) larvae using anti-P1 and anti-L1 antibodies that recognize plasmatocytes and lamellocytes, respectively (Appendix A). While 97.8% of the normal hemocytes accounted for plasmatocytes, no lamellocytes were found among the 1257 cells in the hemolymph (Appendix A). In contrast, 77.4% and 15.0% of the hemocytes in the *mxc^mbn1^* hemolymph were plasmatocytes and lamellocytes, respectively (*n* = 1107 cells) (Appendix A). The statistical analysis suggested that the differences of genotype between *w* and *mxc^mbn1^* influenced the proportion of these two types of hemocytes in the larval hemolymph (Fisher’s exact test; *p* = 3.11 × 10^−5^). Lamellocytes, which were absent in the uninfected controls, were abnormally differentiated in the mutant larvae. Next, we examined whether the hemocytes expressing the undifferentiated cell marker were present in the mutant hemolymph. Mature hemocytes present in the cortical zone of the LG and the circulating hemocytes express the marker gene, *Hml*. In contrast, immature hemocyte precursors in the medullary zone of LGs express *upd3* [26]. We investigated whether circulating hemocytes in the control and mutant hemolymph expressed these two marker genes using *upd3-GAL4* and *Hml-Gal4* drivers combined with *UAS-GFP* (Figure 1a’’’–d’’’). In the normal controls, a GFP fluorescence, which indicated *Hml* expression, was observed in 96.7% of the circulating hemocytes (Figure 1a’’’,e). The fluorescence indicating *upd3* expression was observed in only 3.3% of the normal circulating hemocytes (Figure 1c’’’,e). Since the two percentages add up to 100%, no hemocytes expressing both genes simultaneously were present in the hemolymph of normal larvae (*n* = 1093 cells). In contrast, we observed that the mutant hemocytes showed an *Hml*-dependent GFP fluorescence less frequently (73.6%) than normal hemocytes (Figure 1b’’’,e). The circulating hemocytes expressing *upd3* were observed at a higher frequency (80.4%) in the mutant hemolymph (Figure 1d’’’,e). The evidence that the percentage of cells expressing each gene added together exceeded 100% suggested that hemocytes expressing both *Hml* and *upd3* were present in the *mxc^mbn1^* mutant hemolymph. The statistical analysis suggested that the differences of genotype between *w* and *mxc^mbn1^* influenced the proportion of circulating hemocytes expressing *Hml* and *upd3* (Fisher’s exact test; *p* = 4.44 × 10^−16^). Next, we performed an anti-P1 immunostaining of mutant hemocytes (*mxc^mbn1^/Y*; *upd3>GFP*). Consequently, 85.6% (*n* = 320/374) of the plasmatocytes expressed *upd3* (Appendix A), although no cells expressing the gene were found among all the hemocytes examined in the normal hemolymph (>1000 cells) (Appendix A). Thus, we concluded that over 80% of the differentiated plasmatocytes in the mutant hemolymph expressed *upd3*, a marker of undifferentiated LG cells. These data are consistent with our current conclusion that some mutant hemocytes exhibit transformed phenotypes including defects in cell differentiation.

### 3.2. Hyperactivation of the JAK/STAT Pathway in mxc^mbn1^ Mutant Larvae

Next, we investigated whether the innate-immune-system-related pathway mediated by JAK/STAT was activated in the *mxc^mbn1^* mutant larvae. For this purpose, we first used the GFP reporter, *Stat92E-GFP*, to monitor the pathway activation. We observed an intense GFP fluorescence throughout the body of mutant larvae (*mxc^mbn1^/Y*; *Stat92E-GFP/+*) (Figure 2b’), while a detectable fluorescence was not observed in the normal larvae (*w/Y*; *Stat92E-GFP/+*) (Figure 2a’). The most intense fluorescence was observed in the LG of the mutant larvae (*n* = 10) (Figure 2f’), whereas a fluorescence was not observed in the normal control lobes (*n* = 15) (Figure 2d’). An intense GFP fluorescence was also observed in the anterior region of the fat body in every *mxc^mbn1^* larva (*n* = 20) (Figure 2e’). In contrast, a fluorescence was not observed in normal fat body (Figure 2c’). Thus, we examined the circulating hemocytes derived from LGs and observed an intense fluorescence in approximately 90% of the hemocytes (*n* = 927 cells from 20 larvae) in the *mxc^mbn1^* larvae (Figure 2h”), while we did not observe any GFP fluorescence in the normal hemocytes (*n* = 1575 cells from 20 larvae) (Figure 2g”). Based on these observations, we conclude that the JAK/STAT pathway is activated in tumorous LGs, circulating hemocytes, and fat body of the *mxc^mbn1^* mutant larvae.

### 3.3. Remarkably Increased mRNA Levels of Turandot (Tot) Genes in mxc^mbn1^ Mutant Larvae

As a member of the *Tot* gene family, *TotA* is known to be induced by bacterial infection as well as other stresses via the JAK/STAT signaling pathway [16]. Hence, we investigated whether the mRNA levels of several *Tot* genes were raised in the *mxc^mbn1^* larvae harboring the LG tumor. For this, we decided to quantify the mRNA levels of the four *Tot* genes expressed at the larval stage, *TotA*, *TotB*, *TotC*, and *TotF* (Flybase: FBgn0038838, FBgn0044810, FBgn0044811, and FBgn0028396, respectively), from among the eight known *Tot* family genes. We performed qRT-PCR using total RNAs prepared from the fat body of normal control and *mxc^mbn1^* larvae at the third instar stage (Figure 3). Surprisingly, compared with that in the control larvae (*w/Y*), the mRNA levels of *TotB* and *TotC* increased by >100-fold on average in the 3rd-instar larvae of the *mxc^mbn1^* mutant, respectively (Figure 3b,c) (*p* < 0.05, and not significant with *p* = 0.094, Welch’s *t*-test, respectively), as compared to those of the control larvae (*w^1118^*/*Y*). Consistently, the average mRNA levels of *TotA* and *TotF* were 14- and 60-fold higher, respectively, than those in the control (*p* < 0.01 and *p* < 0.05, Welch’s *t*-test) (Figure 3a,d). Hence, it could be concluded that the mRNA levels of these four *Tot* gene were markedly raised in the fat body of *mxc^mbn1^* larvae. Tot protein expression is also induced by cellular stresses, including viral RNA infection [11]. We excluded the possibility that these genes were induced in response to dsRNAs rather than LG tumors. Thus, we examined the fat-body-specific expression of dsRNA against the *GFP* mRNA in normal larvae (*w/Y*; *r4>GFPRNAi*) and confirmed that the expression of exogenous dsRNA failed to induce the *TotB* or *TotF* mRNAs (Appendix A).

### 3.4. JAK/STAT-Dependent Induction of Gene Expression of Four Tot Genes in the Fat Body of mxc^mbn1^ Larvae

Next, we confirmed whether the marked induction of the four *Tot* genes in the *mxc^mbn1^* larvae was dependent on the JAK/STAT pathway. To achieve this, we induced the ectopic expression of dsRNA against *Stat92E* mRNA using the *UAS-Stat92ERNAi* stock, which efficiently silenced the mRNA in *mxc^mbn1^* larvae (*mxc^mbn1^/Y*; *r4>Stat92ERNAi*). We performed quantitative real-time PCR (qRT-PCR) to measure the mRNA levels of the four *Tot* genes using total RNA isolated from the fat body of mutant larvae at the third instar stage. Consequently, the mRNA levels of all four *Tot* genes decreased significantly by 3.9% of the *TotA* level in *mxc^mbn1^* larvae, 1.4% for *TotF*, <0.2% for *TotB*, and 1.2% for *TotC* in *mxc^mbn1^* larvae with a fat-body-specific silencing of *Stat92E* (Figure 4a–d). The results of the two-way analysis of variance (ANOVA) suggested that there was a significant relationship between the *TotA* mRNA level and *Stat92E* depletion (F(1,8) = 248.9, *p* = 2.60 × 10^−7^), between the *TotA* mRNA level and the genotype (*w* and *mxc^mbn1^*) (F(1,8) = 236.0, *p* = 3.20 × 10^−7^), and their interaction (F(1,8) = 213.6, *p* = 4.71 × 10^−7^). Consistently, there was a significant relationship between the *TotB* mRNA level and *Stat92E* depletion (F(1,8) = 478,650, *p* = 2.13 × 10^−20^), between the *TotB* mRNA level and the genotype (*w* and *mxc^mbn1^*) (F(1,8) = 4478,626, *p* = 2.13 × 10^−20^), and their interaction (F(1,8) = 4476,669, *p* = 2.17 × 10^−20^. There was also a significant relationship between the *TotF* mRNA level and *Stat92E* depletion (F(1,8) = 367.4, *p* = 5.69 × 10^−8^), between the *TotF* mRNA level and the genotype (*w* and *mxc^mbn1^*) (F(1,8) = 323.0, *p* = 9.42 × 10^−8^), and their interaction (F(1,8) = 315.0, *p* = 1.04 × 10^−7^). With regard to the *TotC* mRNA level, there was a statistical significance between the mRNA level and the genotype (*w* and *mxc^mbn1^*) (F(1,8) = 220.69, *p* = 0.002). There was a tendency between the *TotC* mRNA level and the *Stat92E* depletion, although a significant relationship was not observed (F(1,8) = 44.836, *p* = 0.059) possibly due to a larger variation in the mRNA level. There was also an interaction between the *Stat92E* depletion and the genotype (*w* and *mxc^mbn1^*) (F(1,8) = 5.296, *p* = 0.050). In contrast, there was no significant relationship between the *TotC* mRNA level and *Stat92E* depletion (F(1,8) = 44.836, *p* = 0.059), nor in the interaction between the *Stat92E* depletion and the genotype (*w* and *mxc^mbn1^*) (F(1,8) = 5.296, *p* = 0.050). However, there was a statistical significance between the *TotC* mRNA level and the genotype (*w* and *mxc^mbn1^*) (F(1,8) = 220.69, *p* = 0.002). In contrast to the mRNA levels of these *TotA*, *B*, and *F* genes not decreasing significantly in the control larvae harboring a fat-body-specific depletion of *Stat92E* (*w/Y*; *r4>Stat92ERNAi*), the decrease in the mRNA levels of these genes in *mxc^mbn1^/Y*; *r4>Stat92ERNAi* larvae was statistically significantly. Based on these results, we concluded that the expression of these three *Tot* genes was induced by the JAK/STAT pathway in LG tumor mutant larvae.

### 3.5. Silencing TotA, TotB, or TotF Gene by Expressing dsRNAs against the Relevant mRNAs in the Fat Body Enhanced LG Hyperplasia in mxc^mbn1^ Larvae

Genetic data showing that the JAK/STAT pathway was activated in the fat body of *mxc^mbn1^* larvae allowed us to expect that the Tot proteins possessed antitumor property that suppressed LG tumor growth. We investigated the silencing effects of three *Tot* genes, *TotA*, *TotB*, *and TotF*, on the tumor growth in the *mxc^mbn1^* larvae, because the mutant larvae harboring a fat-body-specific silencing of *TotC* had a lower viability. To silence *TotA*, *TotB*, and *TotF* in the mutant fat body, we used *UAS-RNAi* stocks that efficiently silenced the relevant genes. We then measured the whole lobe area of each LG from *mxc^mbn1^* larvae at the mature third instar stage, as described in the Materials and Methods. In contrast to the average LG size in the mutant larvae (*mxc^mbn1^*/*Y*), which was 0.23 μm^2^ (*n* = 21) (Figure 5b,i), the average LG size in the mutant larvae harboring a fat-body-specific *TotB* depletion (*mxc^mbn1^/Y*; *r4>TotBRNAi^1^* and *mxc^mbn1^/Y*; *r4>TotBRNAi^2^*) (0.70 μm^2^ and 0.73 μm^2^, respectively (*n* = 29)) was three times larger than that of *mxc^mbn1^*, at the same time after laying eggs. Consistently, the average size of the LG in the mutant larvae harboring a fat-body-specific depletion of *TotA* (*mxc^mbn1^/Y*; *r4>TotARNAi^1^*, *mxc^mbn1^/Y*; *r4>TotARNAi^2^*) was larger than that in the mutant larvae (*mxc^mbn1^/Y*), which was 0.23 μm^2^, *n* = 21 (Figure 5b,i); (0.6 μm^2^, *n* = 26; and 0.51 μm^2^, *n* = 23, respectively (Figure 5c,d,i). The average LG size of the mutant larvae with *TotF* silencing (*mxc^mbn1^/Y*; *r4>TotFRNAi^1^*, *mxc^mbn1^/Y*; *r4>TotFRNAi^2^*) was also thrice (0.76 μm^2^ and 0.94 μm^2^, respectively (*n* > 20)) that of *mxc^mbn1^* (Figure 5g–i). A one-way ANOVA of the LG size with genotype revealed significant differences (F(7190) = 102.5, *p* = 4.66 × 10^−61^). Subsequently, by using a one-way ANOVA followed by Bonferroni’s multiple comparison test, we confirmed that there were significant differences in the LG size between *mxc^mbn1^* and those harboring the *Tot* RNAi in every six combinations (*p* < 0.0001) (Figure 5i). Thus, the depletion of these three *Tot* genes in the fat body commonly resulted in an enhanced hyperplasia of LG tumors in mutant larvae.

### 3.6. Intake of the Tot B and F Proteins Produced in the Fat Body into the Circulating Hemocytes in mxc^mbn1^ Larvae

Previous studies have demonstrated that three AMPeps expressed in the fat body are transferred to circulating hemocytes and are transported toward tumors (See Introduction). To understand the molecular mechanisms underlying the effects of Tot proteins from the fat body on LG tumors, we selected Tots B and F among the four Tot proteins, since they were more effective than the others. We investigated whether these two proteins induced in the fat body were transported into the circulating hemocytes of the mutant larvae. To visualize the localization of the Tot proteins, we induced the ectopic expression of hemagglutinin (HA)-tagged TotB or TotF using a fat-body-specific *r4-*Gal4 driver and performed anti-HA immunostaining of the hemocytes. We observed a distinctive immunostaining signal for TotB in the cytoplasm of 86.9% of the circulating hemocytes in *mxc^mbn1^* larvae (*n* = 469 cells out of 540 cells) (Figure 6b”), whereas we did not find a distinctive signal above the background level in control hemocytes (*w/Y*; *r4>TotB-3HA*) (*n* = 371 cells) (Figure 6a”). Consistently, 85.5% of the circulating hemocytes in *mxc^mbn1^* larvae showed an anti-HA immunostaining signal for TotF (*n* = 94 cells out of 110 cells) (Figure 6d”), whereas a signal above the background level was not observed in the control hemocytes (*w/Y*; *r4>TotF-3HA*) (*n* > 100 cells) (Figure 6c”). The statistical analysis suggested that the differences in genotype between *w* and *mxc^mbn1^* influenced the proportion of Tot protein uptake in larval hemolymph (Fisher’s exact test, TotB; *p* = 2.28 × 10^−27^, TotF; *p* = 9.44 × 10^−37^). The lower *p*-values supported our conclusion that there were much more circulating hemocytes in mxcmbn1 larvae, which contained the Tot B and F produced in the fat body than in control hemocytes. Higher-magnification images of the hemocytes immunostained with the anti-HA antibody revealed that both proteins were localized within multiple vesicles of various sizes in the cells (see magnified images in the insets of Figure 6b”,d”). We further observed the mutant hemocytes harboring TotF-HA under confocal microscopy by altering the focus planes along the Z-axis. The vesicles stained with HA antibody were present on the confocal planes at a similar depth to those at which the nucleus was localized in the cell’s interior (vertical arrow in Appendix A’). These observations suggested that vesicles containing the Tot proteins were incorporated in the cytoplasm rather than associated with the surface of the mutant hemocytes.

### 3.7. More Hemocytes Containing TotF Proteins Were Associated with the LG Tumors in mxc^mbn1^ Larvae

To test the possibility that the Tot proteins harboring antitumor effects were transported toward the LG tumor via circulating hemocytes, we investigated whether hemocytes containing TotF were localized to the fat body in *mxc^mbn1^* larvae. We performed anti-HA immunostaining of the circulating hemocytes associated with the LGs, instead of immunostaining with the P1 antibody, which also recognizes the LG tumor. We scored a larger number of hemocytes in the mutant LG of *mxc^mbn1^* larvae, in which 243.0 cells (mean)/mm^2^ were scored (*mxc^mbn1^/Y*; *r4>TotF-HA*) (*n* = 14 larvae) than those (mean of 50.3 cells/mm^2^ of the LG) in the control larvae (*w/Y*; *r4>TotF-HA*) (*n* = 8 larvae) (Appendix A). The calculation indicated that five times more hemocytes were attached to the LG tumor than the control LGs (** *p* < 0.01, Welch’s *t*-test) (Appendix A). It is premature to conclude that the hemocytes containing the TotF proteins in the mutant larvae were preferentially recruited to the LG tumors from these results. We found that three times more circulating hemocytes (55,030 hemocytes/L hemolymph) were present in the mutant hemolymph than in the control hemolymph (average of 18,030 hemocytes/L hemolymph). Therefore, considering this evidence, we simply conclude that more hemocytes were localized on the mutant tumor LG.

### 3.8. Fat-Body-Specific Depletion of Tot Genes Enhanced Apoptosis in mxc^mbn1^ LG Tumor

To further understand the mechanism by which Tot proteins suppress LG tumor growth in *mxc^mbn1^* larvae, we investigated whether *Tot* gene silencing in the fat body stimulated apoptosis. Toward this, we performed an anti-cDcp immunostaining of LGs from the third-instar larvae to detect the activated form of the effector caspase. In the mutant LGs (*mxc^mbn1^/Y*; *r4>+*), an anti-cDcp1 immunostaining signal was observed in 39.5% of the whole lobe areas on average (*n* = 23) (Figure 7b”,f), whereas a subtle signal was observed in a limited area (1.1%) of the normal lobes (*w/Y*) (*n* = 20) (Figure 7a”,f). In contrast, the percentage of the LG areas exhibiting the apoptotic signal significantly decreased by 14.3% of the whole LG lobes on average (*n* = 18) in the mutant larvae harboring a fat-body-specific depletion of *TotB*. The cDcp1-positive areas in *mxc^mbn1^* LGs increased by the *TotB* depletion, while there were no significant differences among the *mxc^mbn1^* larvae harboring fat-body-specific *TotBRNAi*. Consistently, the average percentage of the LG area exhibiting an apoptotic signal decreased by 18.3% of the whole LG lobes (*n* = 18) in the mutant larvae harboring a fat-body-specific silencing of *TotF* (*n* = 15) (Figure 7e”,f). A one-way ANOVA of Dcp1 area’s proportion with the genotype as the effect revealed significant differences (F(4,95) = 31.26, *p* = 1.32 × 10^−16^) (Figure 7d”,f). A subsequent one-way ANOVA followed by Bonferroni’s multiple comparison test did not show significant differences in the cDcp1 areas between *mxc^mbn1^*; *r4>*+ and *mxc^mbn1^*; *r4>TotBRNAi* (*p* = 0.156) or *mxc^mbn1^*; *r4>TotFRNAi* (*p* = 0.716) (Figure 7f). However, when we compared the mean values of *mxc^mbn1^*; *r4>TotBRNAi* with that of *mxc^mbn1^/Y*, there was a significant difference (*p* = 0.005) but not for *mxc^mbn1^*; *r4>TotFRNAi* (*p* = 0.082). As the mean proportions of the apoptosis areas in both cases were lower than the control proportion, these results suggested a possibility that the silencing of *TotB* and *TotF* in the fat body may reduce apoptosis induction in *mxc^mbn1^* LGs.

In addition, we observed a bimodal distribution of the proportion of apoptosis areas that were detected by anti-cDcp1 immunostaining in the LGs of *mxc^mbn1^* larvae (*mxc^mbn1^* and *mxc^mbn1^*; *r4>*+) (Figure 7f). A similar bimodal distribution was also observed in the mutant larvae harboring a fat-body-specific depletion of *TotB* or *TotF*. This distribution resulted in a larger standard deviation of apoptosis areas in LGs for each genotype and thereby, in no statistically significant differences between *mxc^mbn1^* and the mutant harboring the *Tot* depletion (Figure 7f).

### 3.9. RNAi-Based Silencing of Tot Genes Also Increased Mitotic Cells in the LGs of mxc^mbn1^ Larvae

To further understand why tumor growth was enhanced upon silencing of *TotB* or *TotF*, we investigated whether the ectopic expression of Tot proteins affected the proliferation of LG cells in the mutant larvae. First, we visualized mitotic cells by means of immunostaining with an anti-pH3 antibody in the LGs of *mxc^mbn1^* larvae harboring a fat-body-specific silencing of *TotB* (*mxc^mbn1^/Y*; *r4>TotBRNAi*) and *TotF* (*mxc^mbn1^/Y*; *r4>TotFRNAi*), and then counted the number of mitotic cells in the lobe areas of the mutant LGs with and without a fat-body-specific silencing of the *Tot* genes (Figure 8). We scored phospho-H3 (pH3)-positive cells and converted the cell numbers to 0.2 cells on average (*n* = 15) per constant area (1 μm^2^) of the *mxc^mbn1^* LGs (*mxc^mbn1^/Y*; *r4>+*) (Figure 8c,f). In contrast, we observed three times more pH3-positive cells in the LGs of the mutant larvae harboring a fat-body-specific depletion of *TotB* and *TotF* (0.63 and 0.55 cells on average in the same LG areas (*n* = 15, 20, respectively) (Figure 8d–f). A one-way ANOVA of the pH3-positive cell number with the genotype as the effect suggested significant differences F(4,99) = 21.27, *p* = 1.11 × 10^−12^ (Figure 8f). A subsequent one-way ANOVA followed by Bonferroni’s multiple comparison test confirmed that there were significant differences in the numbers of pH3-positive cells between *mxc^mbn1^*; *r4>*+ and *mxc^mbn1^*; *r4>TotBRNAi* (*p* < 0.001) or *mxc^mbn1^*; *r4>TotFRNAi* (*p* < 0.001) (Figure 8f). These observations indicated that the mitotic cells in LG tumors were increased by the downregulation of *Tot* genes in the mutant fat body3 and suggested that these Tot proteins also suppressed the proliferation of tumor cells in *mxc^mbn1^* larvae.

We also observed a bimodal distribution of the number of mitotic cells in the LGs of *mxc^mbn1^* larvae harboring fat-body-specific depletion of TotB (*mxc^mbn1^*; *r4>TotBRNAi*), but not in the mutant larvae. This distribution resulted in a relatively larger standard deviation of mitotic cells in LGs (Figure 8f). The clearly bimodal distribution was not observed in *mxc^mbn1^*; *r4>TotFRNAi*.

## 4. Discussion

In this study, we investigated whether Turandot (Tot) family polypeptides, which are induced in response to microbial infection, exert antitumor effects against *Drosophila* hematopoietic tumors in *mxc^mbn1^* mutant larvae. Moreover, we addressed the mechanism of how the proteins suppressed the tumor growth. First, we found that many abnormal hemocytes expressing *upd3*, a marker for undifferentiated LG cells, were present in the mutant hemolymph. Next, we showed that the JAK/STAT pathway downstream of the ligand was activated in the mutant larvae. We further demonstrated that the activation of the pathway induced the transcription of its target genes encoding Tot proteins in the mutant larvae. Subsequently, Tot proteins exerted antitumor effects by suppressing cell proliferation and inducing apoptosis in the tumor.

### 4.1. Activation of the JAK/STAT Pathway by Ectopic Expression of Upd3 from Macrophage-like Cells and Subsequent Induction of Tot Proteins in the Fat Body

The ectopic expression of Upd3 was observed in macrophage-like circulating hemocytes of *mxc^mbn1^* larvae harboring hematopoietic tumors, whereas this expression was negligibly observed in their normal counterparts. Consistently, an epigenetic misexpression of genes encoding Upd ligands has been reported in *Drosophila* tumors [33,34]. Interleukin-6, a functional counterpart of Upd3 in mammals, is also induced; accordingly, the activation of the JAK-STAT pathway is observed in several tumors, such as breast and ovarian cancer [35,36]. In this study, we demonstrated that the JAK/STAT pathway was activated in mutant hemocytes, and that the mRNA levels of the *Tot* genes increased in the mutant fat body. Because Upd3 is a ligand for the JAK/STAT pathway [37,38], it is continuously activated in LG-derived hemocytes in hematopoietic tumors [39]. The expression of Upd cytokines is promoted by the activation of JNK signaling [13,40,41]. We recently demonstrated that the JNK pathway was activated in mutant LGs and circulating hemocytes (Takarada and Inoue, submitted). The JNK activation eventually results in the induction of Upd cytokines [13,40,41]. Therefore, we speculated that hemocytes in which the JNK pathway was activated expressed Upd3 in mutant larvae, and consequently, the JAK/STAT pathway was activated in larval cells and tissues.

Some types of *Drosophila* tumor cells, including hemocytes in *mxc^mbn1^* larvae, accumulate reactive oxygen species (ROS) at high levels, and simultaneously, are associated with an increased expression of matrix metalloproteinases (Mmp1 and 2) [22,23,42]. The overexpression of Mmps stimulates the degradation of the basement membrane in tissues. Conversely, the inhibition of Mmps in *Drosophila* tumor tissues suppresses tumor invasion [43]. Considering these previous findings, it is possible to speculate that circulating hemocytes may be involved in the recognition of the LG tumor in the mutant larvae, for example, by recognizing a higher Mmp expression or the resultant loss of tissue integration. These tumor phenotypes can promote Upd3 expression via JNK activation [13]. Our observation that more hemocytes were associated with LG tumors and that the JAK/STAT pathway was activated in the fat body in the mutant larvae let us speculate that the hemocytes might migrate toward the fat body while secreting Upd3. The secreted ligand then binds to its receptors in the fat body, thereby activating the JAK/STAT pathway. Consequently, Tot proteins, encoded by the target genes of the JAK/STAT pathway, are produced and secreted from the fat body. The silencing of *upd3* in hemocytes inhibits the production of the TotA protein [44]. We showed that more circulating hemocytes containing TotB and TotF proteins were associated with tumorous LG in the mutant larvae than in the normal LGs, suggesting that the hemocytes may be preferentially recruited to the LG tumors. Kinoshita and colleagues recently reported that normal hemocytes were selectively recruited to the LG tumors [23]. They showed that the induction of three AMPeps, Drs, Def, and Dip, in *mxc^mbn1^* larvae required circulating hemocytes to produce high levels of reactive oxygen species (ROS) at a higher level. The authors argued that ROS from hemocytes contributed to the activation of Toll in the innate immune pathway, which stimulated AMPep production [23]. As there is yet no evidence that ROS are involved in JAK/STAT activation, the underlying signaling pathway could be activated via a different mechanism. To verify our model, it is important to determine whether Upd3 from hemocytes is directly involved in the activation of the JAK/STAT pathway in the fat body.

### 4.2. Tot Proteins Possess Antitumor Property That Induced Apoptosis in the Hematopoietic Tissue Tumors in a Drosophila Model of Hematopoietic Tumor

We showed that the depletion of three *Tot* genes in the fat body suppressed the induction of apoptosis and increased the number of mitotic cells in the LG tumors. These results suggested that Tot proteins exert their antitumor effects by inducing apoptosis and inhibiting tumor cell proliferation. Although the proteins themselves are considered cytotoxic, no growth inhibition was observed in the fat body and LGs of normal control larvae harboring the fat-body-specific induction of TotB and TotF. Most of the circulating hemocytes did not take up these proteins in the control larvae, in contrast to the hemocytes in the *mxc^mbn1^* larvae. Thus, the growth-inhibitory effect of the proteins exclusively on tumor cells may primarily involve in the tumor recognition by circulating hemocytes. Reports have shown that AMPeps generated via the innate immune pathways mediated by Toll and Imd exert anti-tumor effects [7,8]. These AMPeps are originally induced after infection with Gram-negative bacteria, Gram-positive bacteria, or both [3,4]. Some proteins contain cationic and amphiphilic amino acid sequences that are helpful for binding to the bacterial cell surface, thereby forming pores and disrupting the cell membrane structure [45]. *Drosophila* AMPeps can destroy invading microorganisms and suppress the progression of several types of tumors in larvae [7,8,46]. Araki and colleagues reported that the overexpression of five AMPeps enhanced apoptosis in *mxc^mbn1^* LG tumors, whereas no apoptotic signals were detected in the controls. AMPeps contain positively charged amino acid sequences that target negatively charged microbial membranes [47,48]. This electrostatic interaction may also be effective when AMPeps act on tumors, and tumors arising from imaginal discs of *dlg* mutants contain phosphatidylserine (PS) exposed on the cell membrane surface. This phospholipid is negatively charged, and one of the AMPeps, defensin, may recognize the phosphatidylserine exposed on the cell surface [8]. Similar electrostatic interactions may be involved in the specific cytotoxic effects of Tot proteins on tumor cells. The TotA protein is an acidic and highly charged 12 KDa polypeptide [16]. Possibly, TotB and TotF proteins also exhibit antitumor properties that induce apoptosis via a mechanism similar to that proposed for those of the AMPeps. In addition, as TotA and TotM are generated in response to multiple cellular stressors [49], we cannot exclude the possibility that apoptosis could be induced as a consequence of the cellular stress caused by the proteins.

### 4.3. Another Antitumor Effect of the Tot Proteins due to the Suppression of LG Tumor Cell Proliferation

In addition to the antitumor effect that induces apoptosis in the tumors, we also showed that the depletion of *TotB* or *TotF* resulted in threefold or more increases in the number of proliferating cells in the tumors of *mxc^mbn1^* larvae. These genetic data suggested that the Tot proteins exhibited antitumor property by inhibiting cell proliferation in the tumors. In contrast, neither defensin nor diptericin suppress the proliferation of tumor cells, while both proteins induce cell death in LG tumors [7]. The influence of the insulin/insulin-like growth factor (IGF) system on cell growth and proliferation plays a critical role in the acquisition of a malignant phenotype and the progression of several cancer cells [50,51,52]. The biological activity of IGFs depends on their binding to IGF receptors and their interaction with the IGF-binding protein (IGFBP) family [50]. A *Drosophila* orthologue of IGFBP7, Ecdysone-inducible gene L2 (ImpL2), antagonizes *Drosophila* insulin-like peptides essential for cell growth and metabolism to induce the wasting phenotype in tumor-hosting adults [53]. It is also possible that Tot proteins antagonize growth factor(s) that stimulate LG cell proliferation, similar to the IGBP family of proteins. Therefore, it would be interesting to further investigate the antitumor effects of Tot proteins on LG tumor in future.

### 4.4. Possible Origin of Bimodal Distribution of Apoptosis and Mitotic Cells in LGs of mxc^mbn1^ and the Mutant Larvae Harboring Fat-Body-Specific Tot Depletion 

A clearly bimodal distribution of the apoptosis area was observed in the LGs of *mxc^mbn1^* and the mutant larvae harboring a fat-body-specific depletion of *TotB* or *TotF*. We also noticed a bimodal distribution of mitotic cells in LGs of the mutant larvae harboring fat-body-specific *TotB*RNAi, although the bimodality was exhibited less apparently. Even in the LGs of individual mutant larvae at almost the same development stage, the size varies widely among each individual larvae [7,21]. Not all LG cells were transformed in the mutant larvae, and the frequencies of the tumor cells varied among larvae, consistent with the common nature known as intratumor heterogeneity [7,22,54]. The bimodal distributions allow us to speculate a possible threshold in the induction or action of Turandot. When an LG size exceeds a certain size limit, this may trigger the induction of the canonical innate immune pathways and JAK/STAT pathway in the fat body. When Gram-negative bacteria are infected in *Drosophila* adults deficient in the *Toll* or phagocytosis gene, a similar bimodal distribution of infection outcomes persists. The observed divergent infection is considered to be a natural result of mutual negative feedback between pathogens and the host immune response [55]. Additionally, a substantial variation in mRNA levels of the AMP genes that are targets of the Toll- and/or Imd-mediated pathways has been observed after bacterial infection or in *mxc^mbn1^* larvae [7,23,55]. Theoretical analyses have suggested that small differences among hosts can be magnified into life-or-death differences to create a bimodality in infection outcomes [56]. Alternatively, as we discussed above, if Tot B and F succeed in arresting the cell cycle of LG tumor cells, some of the cells that have been inhibited from proliferation may have escaped from apoptosis. The two subpopulations of tumor cells, which harbor a stronger and weaker immunostaining signal for activated effector caspase, may correspond to the LG cells undergoing apoptosis and those assured survival, respectively. The LG cells exhibiting a stronger apoptosis signal might have been closely associated with the circulating hemocytes containing TotB or TotF and received the antitumor proteins. Further experiments need to be performed to clarify these interpretations. 

### 4.5. Contribution of the Innate Immune System to Suppression of Tumor Growth in Drosophila

Most genes involved in the *Drosophila* innate immune system are highly conserved in mammals [1,57]. The *Drosophila* innate immune pathways mediated by Toll and Imd are almost equivalent with respect to the components and cascades of the mammalian Toll-like receptor (TLR) and tumor necrosis factor receptor (TNF) signaling pathways, respectively [58,59,60]. The lack of an acquired immune system in *Drosophila* makes it easier to highlight its effects on the innate immune system. Therefore, studies on the *Drosophila* innate immune system could contribute to the elucidation of the human innate immune system. For example, Toll-like receptors (TLRs), the most important proteins in the mammalian innate immune system, have been identified as orthologs of *Drosophila* Toll [1,58]. We hypothesize that mammals may also harbor functional counterparts of the anticancer peptides induced via the JAK/STAT pathway. If this is the case, our current findings will improve the understanding of the mechanism(s) by which these anticancer proteins suppress tumor growth. We propose the following model based on the results of the present study: In *Drosophila* hematopoietic tumors, macrophage-like hemocytes first recognize the tumor and then express the inflammatory cytokine, Upd3. The Upd3-expressing hemocytes may migrate to the fat body while secreting Upd3 in *Drosophila*. The binding of Upd3 to its receptors on the fat body activates the JAK/STAT signaling pathway and induces the transcription of *Tot* genes. Proteins secreted from the fat body are incorporated into circulating hemocytes, which might recognize the tumor and migrate toward it to release the proteins. The released Tot protein inhibits tumor growth and induces apoptosis, thereby exerting antitumor effects. Additionally, we are currently unable to exclude the possibility that the mutant hemocytes adhering to the fat body invaded the other tissues rather than transmitted the information, although normal hemocytes transplanted into the *mxc^mbn1^* larvae are preferentially recruited to the LG tumors [23]. However, there is a need for further studies to validate this model. To investigate the depletion effect of each *Tot* gene, we used the *UAS-RNAi* stocks that did not induce any expression of off-target genes [30]. However, we cannot completely exclude the possibility that simultaneous nonspecific downregulation of other *Tot* genes or nonrelated genes would be involved in the effect. To strengthen our model, it would be important to investigate whether the ectopic expression of whole cDNA, in which the sequences used for the depletion are not contained, cancels the depletion effect of the *Tot* genes.

## 5. Conclusions

The major findings in this study are as follows: (1) the *upd3* expression was induced in the hemocytes of *Drosophila mxc* mutant larvae harboring LG tumors, and consequently, the JAK/STAT pathway was activated in the mutant fat body; (2) a family of proteins called Tots, encoded by target genes of this signaling pathway, was induced; (3) two members of the family, TotB and TotF, were taken up only in the mutant hemocytes, and the cells were closely associated with the LG tumors; (4) these Tot proteins exhibited antitumor effects that suppressed tumor growth via the inhibition of tumor cell proliferation and the induction of apoptosis.

## Figures and Tables

**Figure 1 cells-12-02047-f001:**
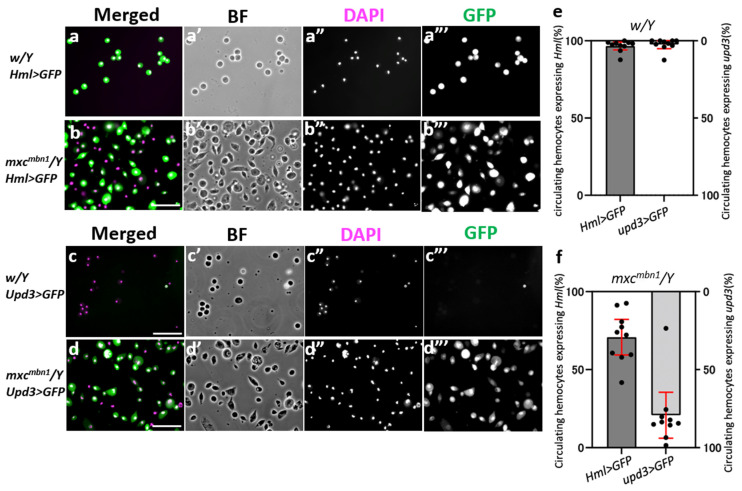
Increased frequencies of circulating hemocytes expressing *upd3*, a marker for undifferentiated hemocyte precursor, in *mxc^mbn1^* mutant larvae. (**a**,**b**) Fluorescent images of the hemocytes expressing GFP depending on *Hml* in normal control (*w/Y*; *Hml>GFP*) (**a**), and *mxc^mbn1^* (*mxc^mbn1^/Y*; *Hml>GFP*) larvae (**b**). (**a’**,**b’**) Bright-field images (BF) of the hemocytes: the GFP fluorescence signal in green in (**a**,**b**) (white in (**a’’’**,**b’’’**)), while DNA is stained in magenta in (**a**,**b**) (white in (**a”**,**b”**)). (**c**,**d**) Fluorescence images of the hemocytes expressing GFP depending on *Upd3* in normal control (*w/Y*; *upd3>GFP*) (**c**) and *mxc^mbn1^* (*mxc^mbn1^/Y*; *upd3>GFP*) (**d**) larvae. Bright-field images of the hemocytes (**c’**,**d’**): the GFP fluorescence signal is in green in (**c**,**d**) (white in (**c’’’**,**d’’’**)), while DNA is stained in magenta in (**c**,**d**) (white in (**c”**,**d”**)). Scale bar: 10 μm. (**e**,**f**) Graphs quantifying the percentage of the cells expressing GFP depending on *Hml* and/or *upd* in control (*w/Y*) (**e**) or the mutant (*mxc^mbn1^*) (**f**) larvae. Error bars represent the 95% confidence intervals (95%CI). Thus, the *mxc^mbn1^* hemolymph contained hemocytes with a higher frequency of *upd3*, a marker gene for undifferentiated hemocyte precursors.

**Figure 2 cells-12-02047-f002:**
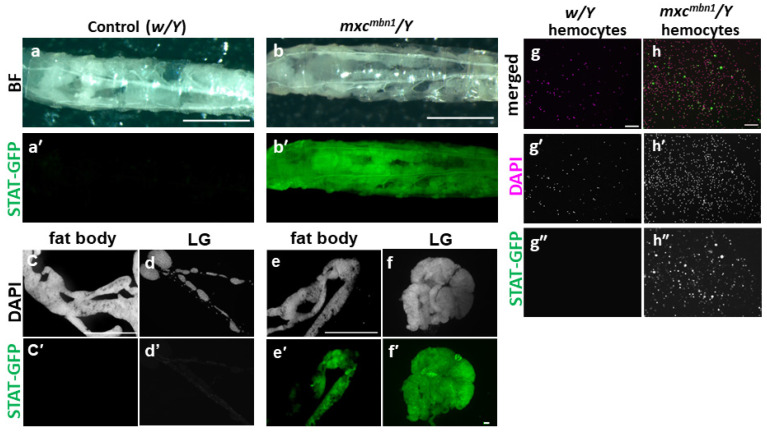
Activation of the JAK/STAT pathway in the fat body, LGs, and circulating hemocytes of *mxc^mbn1^* mutant larvae. (**a**–**b’**) Larval side view, anterior to left. (**a**,**b**) Bright-field (BF) images and (**a’**,**b’**) GFP fluorescence images of mature control (**a**,**a’**) and *mxc^mbn1^* mutant (**b**,**b’**) larvae harboring the *10×Stat92E-GFP* reporter, which monitors the activation of the JAK-STAT pathway. Scale bar is 1 mm. (**c**–**f**) DAPI-stained fluorescence images and GFP fluorescence (**c’**–**f’**) images of the fat body (**c**,**e**) and LGs (**d**,**f**) in normal control (*w/Y*; *Stat92E-GFP/+*) (**c**,**d**) and *mxc^mbn1^* mutant larvae (*mxc^mbn1^/Y*; *Stat92E-GFP/+*) (**e**,**f**). (**g**,**h**) DAPI-stained fluorescence images (magenta in (**g**,**h**), white in (**g’**,**h’**)) and GFP fluorescence (green in (**g**,**h**), white in (**g”**,**h”**)) fluorescence images of the circulating hemocytes in the hemolymph of normal control (*w/Y*; *Stat92E-GFP/+*) (**g**) and *mxc^mbn1^* (*mxc^mbn1^/Y*; *Stat92E-GFP/+*) mutant larvae (**h**). Scale bar: 100 μm.

**Figure 3 cells-12-02047-f003:**
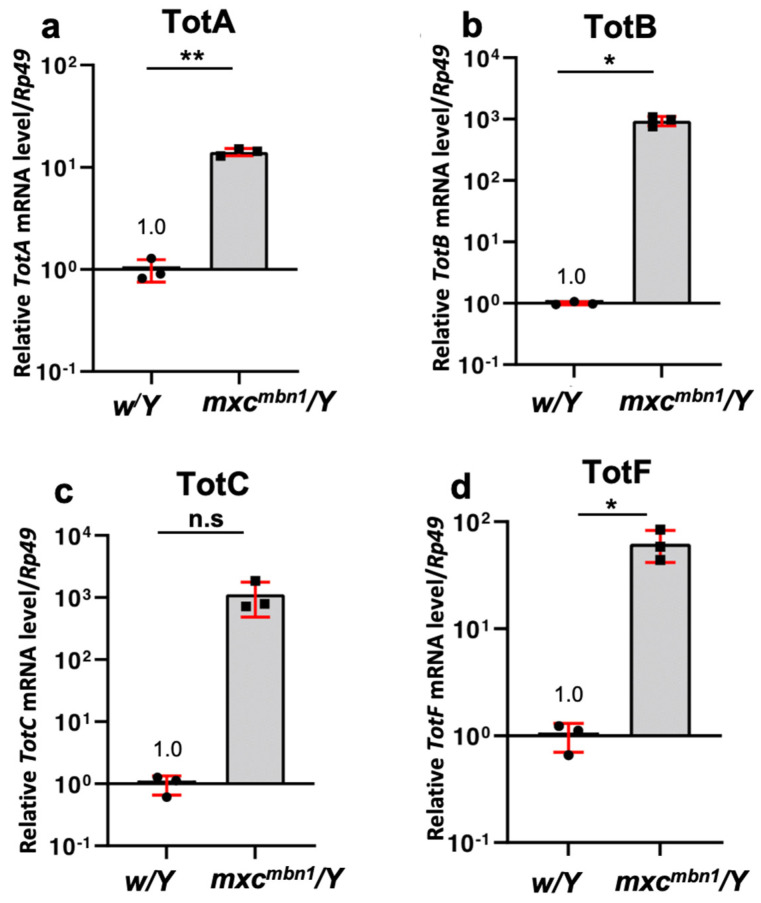
qRT−PCR to quantify the mRNA levels of *TotA*, *TotB*, *TotC*, and *TotF* in *mxc^mbn1^* mutant larvae. (**a**–**d**) Quantification of the mRNA levels of *TotA, TotB, TotC*, and *TotF* using qRT-PCR with total RNAs prepared from third-instar larvae used as templates. The x-axis from left to right shows normal control (*w/Y*) and *mxc^mbn1^* mutant (*mxc^mbn1^/Y*) larvae. The y-axis shows the mRNA levels of the target genes relative to the endogenous control gene, *Rp49*. The mean of the relative value of the control from three independent experiments is shown as 1.0. Bars indicate the relative mRNA levels of the target genes, *TotB* and *TotF*, as assessed using qRT-PCR (average of three times). The differences between groups were assessed using Welch’s *t*-test (TotA t(2.183) = 19.20, *p* = 0.002; TotB t(2.000) = 9.698, *p* = 0.011; TotC t(2.000) = 3.301, *p* = 0.094; TotF t(2.001) = 5.144, *p* = 0.036, * *p* < 0.05 and ** *p* < 0.01, n.s.—not significant, *n* = 3). Error bars indicate standard deviation (s.d.).

**Figure 4 cells-12-02047-f004:**
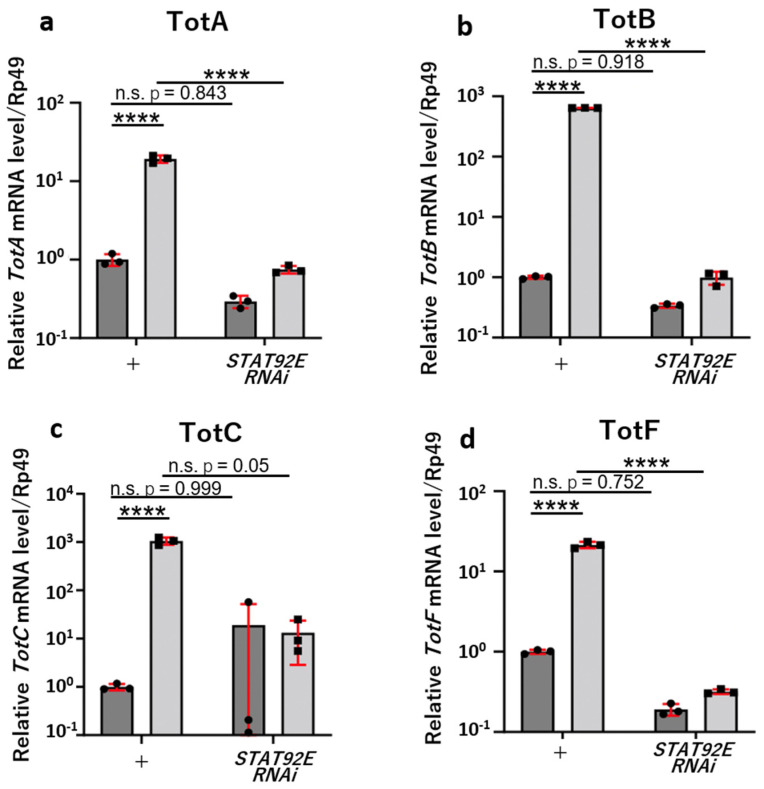
JAK/STAT pathway−dependent increase in the transcription of *Tot* genes in the fat body of *mxc^mbn1^* mutant larvae. (**a**–**d**) Quantification of the mRNA levels of *TotA*, *TotB*, *TotC*, and *TotF* using qRT-PCR, with RNAs prepared from the fat body used as templates. Quantification was performed using qRT-PCR with RNAs prepared from the fat body as templates. The x-axis from left to right: normal control larvae (*w/Y*; *r4/+*), fat-body-specific *Stat92E* depletion in control larvae (*w/Y*; *r4>Stat92ERNAi*), *mxc^mbn1^* mutant larvae (*mxc^mbn1^/Y*; *r4/+*), and *mxc^mbn1^* larvae with fat-body-specific *Stat92E* depletion in (*mxc^mbn1^/Y*; *r4>Stat92ERNAi*). The y-axis shows the relative mRNA levels of the target genes to the level of control gene levels (*Rp49*). The mean of the relative value of the control from three independent experiments is shown as 1.0. Bars indicate the relative mRNA levels of the target gene. Statistical difference was examined using a two-way ANOVA by Turkey’s multiple comparisons tests. **** *p* < 0.0001 Error bars indicate s.d.

**Figure 5 cells-12-02047-f005:**
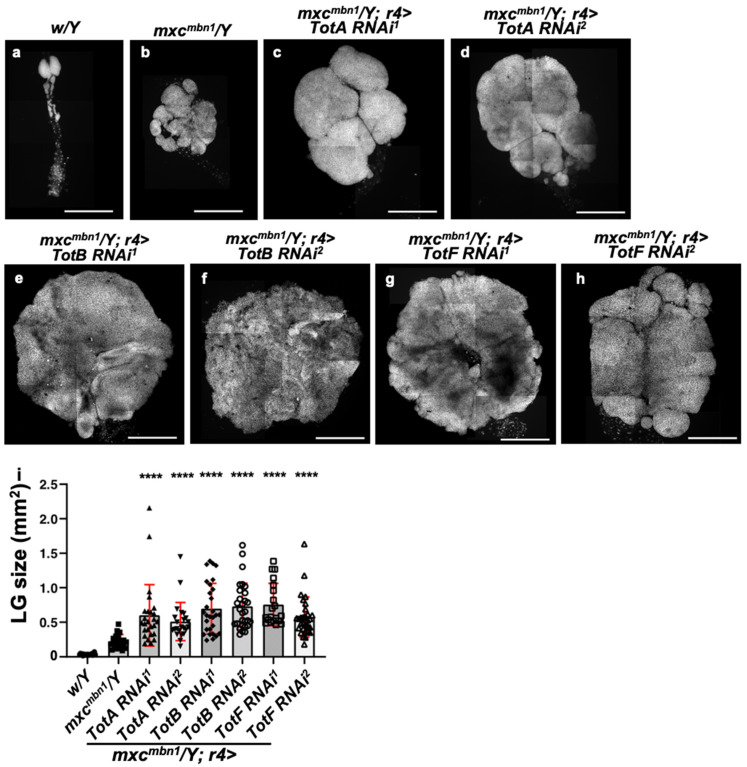
Enhancement of the hyperplasia of LG tumors after depletion of *TotA*, *TotB*, and *TotF* in the fat body of *mxc^mbn1^* mutant larvae. (**a**–**h**) DAPI-stained images of LGs in (**a**) normal control (*w/Y*) larva, (**b**) *mxc^mbn1^* mutant larva without depletion (*mxc^mbn1^/Y*), (**c**,**d**) *mxc^mbn1^* mutant larva harboring a fat-body-specific depletion of *TotA* (*mxc^mbn1^/Y*; *r4>TotARNAi^1^* (**c**) and *mxc^mbn1^/Y*; *r4>TotARNAi^2^* (**d**)), (**e**,**f**) *mxc^mbn1^* mutant larva with *TotB* knockdown (*mxc^mbn1^/Y*; *r4>TotBRNAi^1^* (**e**) and *mxc^mbn1^/Y*; *r4>TotBRNAi^2^* (**f**)), and (**g**,**h**) *mxc^mbn1^* mutant larva with a depletion of *TotF* (*mxc^mbn1^/Y*; *r4>TotFRNAi^1^* (**g**) and *mxc^mbn1^/Y*; *r4>TotFRNAi^2^* (**h**)). Scale bar is 100 μm. (**i**) Quantification of the LG size in larvae with depletion of *TotA*, *TotB*, or *TotF* in the *mxc^mbn1^* larvae. Statistical difference tests were determined using a one-way ANOVA (TotA, *n* = 23; TotB, *n* = 29; TotF, *n* = 20). The *p*-value above each bar indicates the statistical difference between *mxc^mbn1^/Y* and *mxc^mbn1^/Y*; *r4>TotRNAi*, as follows; by using a one-way ANOVA followed by Bonferroni’s multiple comparisons test (*p* = 1.62 × 10^−17^ between control and *mxc* mutants, *p* = 1.83 × 10^−8^ between *mxc* and the mutant with *TotARNAi^1^*, *p* = 1.40 × 10^−6^ between *mxc* and *TotARNAi^2^*, *p* = 1.64 × 10^−12^ between *mxc* and *TotBRNAi^1^*, *p* = 2.6 × 10^−14^ between *mxc* and *TotBRNAi^2^*, *p* = 1.99 × 10^−13^ between *mxc* and *TotFRNAi^1^*, *p* = 1.19 × 10^−9^ between *mxc* and *TotFRNAi^2^*, and **** *p* < 0.0001. Error bars indicate s.d.

**Figure 6 cells-12-02047-f006:**
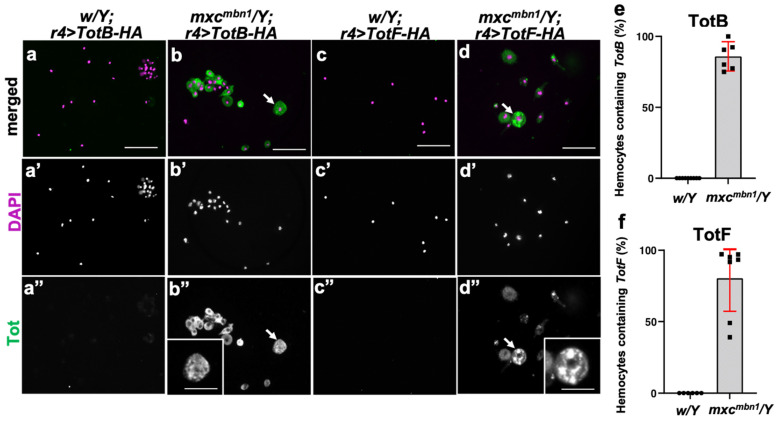
Immunostaining of circulating hemocytes to detect TotB and TotF induced in the fat body of *mxc^mbn1^* and control larvae. (**a**–**d**) Fluorescent images of circulating hemocytes immunostained with anti-HA antibody to detect TotB (**a**,**b**) and TotF (**c**,**d**). (**a**,**b**) normal control (*w/Y*; *r4>TotB-HA*) (**a**) and *mxc^mbn1^* larvae harboring the fat-body-specific overexpression of *TotB* (*mxc^mbn1^/Y*; *r4>TotB-HA*) (**b**). (**c**,**d**) TotF in normal control (*w/Y*; *r4>TotF-HA*) (**c**) and *mxc^mbn1^* larvae harboring the fat-body-specific expression of *TotF* (*mxc^mbn1^/Y*; *r4>TotF-HA*) (**d**). Anti-HA immunostaining signal is in green in (**a**–**d**) (white in (**a”**–**d”**)), while DNA is stained in magenta in (**a**–**d**) (white in (**a’**–**d’**)). Arrows indicate vesicles immunostained with anti-HA antibody in hemocytes. The hemocytes are indicated using arrows in (**b**,**d**), with the magnified forms shown in insets (**b”**,**d”**). The cells containing vesicles were visualized by means of anti-HA immunostaining. Scale bars: 10 μm (**b**,**d**) and 5 μm (**b”**,**d”**). (**e**,**f**) Graphs quantifying the percentages of calculating hemocytes containing TotB (**e**) and those containing TotF (**f**) among total circulating hemocytes in w and *mxc^mbn1^* larvae. Error bars represent 95%CI.

**Figure 7 cells-12-02047-f007:**
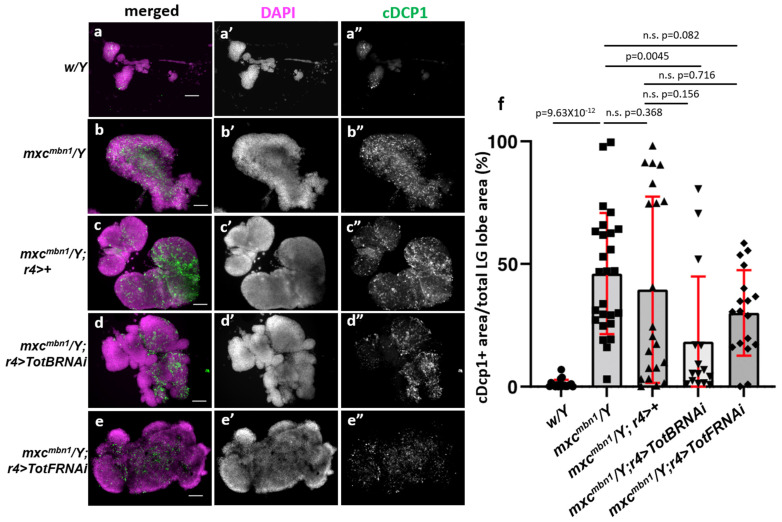
Immunostaining of LGs with the antiactivated caspase antibody to detect apoptotic cells on the LG tumors of *mxc^mbn1^* larvae harboring *TotB* and *TotF*. (**a**–**e**) Fluorescent images of LGs immunostained with an anti-cDcp1 antibody in (**a**) normal control larvae (*w/Y*), (**b**) *mxc^mbn1^* larvae (*mxc^mbn1^/Y*), (**c**) *mxc^mbn1^* larvae expressing Gal4 depending on *r4* (*mxc^mbn1^/Y*; *r4>+*), (**d**) *mxc^mbn1^* larval harboring the fat-body-specific depletion of *TotB* (*mxc^mbn1^/Y*; *r4>TotBRNAi^1^*), and (**e**) the LG of the mutant larvae harboring the depletion of *TotF* (*mxc^mbn1^/Y*; *r4>TotFRNAi^1^*). The anti-cDcp1 immunostaining signal is in green (white in (**a”**–**e”**)), while DNA is stained in magenta in (**a**–**e**) (white in (**a’**–**e’**)). Scale bar: 100 μm. (**f**) Quantification of the area showing anti-cDcp1 immunostaining signal in whole lobe regions of LGs of each genotype. Statistical significance was examined using a one way ANOVA test (*n* ≥ 15). *p* = 9.63 × 10^−12^ control and *mxc*, *p* = 0.368 between *mxc^mbn1^*/Y and *mxc^mbn1^/Y*; *r4>+*, *p* = 0.156 between *mxc^mbn1^/Y*; *r4>+* and *mxc^mbn1^/Y*; *r4>TotBRNAi^1^*, *p* = 0.716 between *mxc^mbn1^*; *r4>+* and *mxc^mbn1^*; *r4>TotFRNAi^1^.* n.s.: not significant. Error bars indicate s.d.

**Figure 8 cells-12-02047-f008:**
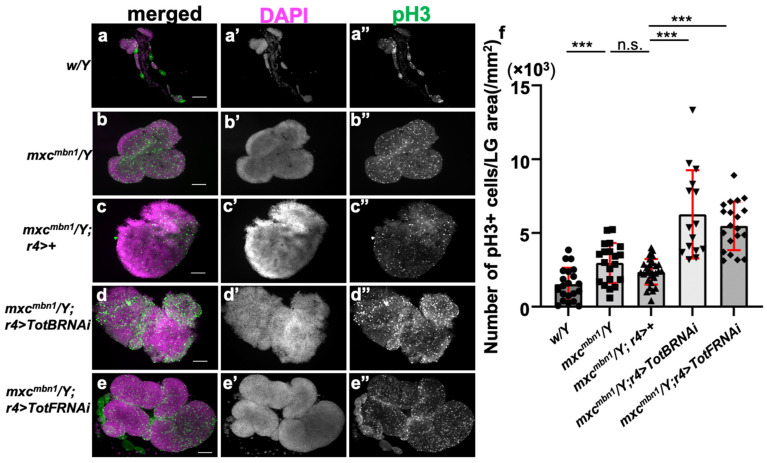
Anti-phospho-H3 immunostaining of LGs, to detect proliferation in the LG tumors of *mxc^mbn1^* mutant larvae harboring a depletion of *TotB* and *TotF*. (**a**–**e**) Fluorescent images of LGs immunostained with anti-pH3 antibody from (**a**) normal control (*w/Y*) larva, (**b**) *mxc^mbn1^* mutant (*mxc^mbn1^/Y*) larva, (**c**) *mxc^mbn1^* mutant larva expressing Gal4 dependent on *r4* (*mxc^mbn1^/Y*; *r4>+*), (**d**) *mxc^mbn1^* mutant larva harboring a fat-body-specific depletion of the *TotB* gene (*mxc^mbn1^/Y*; *r4>TotBRNAi*), and (**e**) *mxc^mbn1^* larva harboring the depletion of *TotF* (*mxc^mbn1^/Y*; *r4>TotFRNAi*). The anti-pH3 immunostaining signal is in green (white in (**a”**–**e”**)), while the DNA is stained in magenta in (**a**–**e**) (white in (**a’**–**e’**)). Scale bar is 100 μm. (**f**) Number of mitotic cells in the LGs from larvae of each genotype. Statistical significance was examined using a one-way ANOVA test followed by Bonferroni’s multiple comparisons test (*n* ≥ 15). *p* = 1.36 × 10^−4^ between control and *mxc*, *p* = 0.244 between *mxc^mbn1^*/Y and *mxc^mbn1^/Y*; *r4>+*, *p* = 2.76 × 10^−4^ between *mxc^mbn1^/Y*; *r4>+*, and *mxc^mbn1^/Y*; *r4>TotBRNAi^1^*, *p* = 2.67 × 10^−4^ between *mxc^mbn1^*; *r4>+*, and *mxc^mbn1^*; *r4>TotFRNAi^1^*. *** *p* < 0.001, n.s.: not significant. Error bars indicate s.d.

## Data Availability

The datasets generated and/or analyzed in the current study are available from the corresponding author on reasonable request.

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
