# Peer review of "Anti-Tumor Effect of Turandot Proteins Induced via the JAK/STAT Pathway in the mxc Hematopoietic Tumor Mutant in Drosophila"

_cells, 2023, doi:10.3390/cells12162047_

Round 1
Reviewer 1 Report
The manuscript by Kinoshita and coauthors concerns with tumor-associated consequences of JAK/STAT-promoted increase in levels of Turandot transcripts in Drosophila multi sex comb (mxc) mutant larvae.
They studied the changes in circulating hemocyte numbers, transcript levels for several Turandot genes and the consequences of RNAi-mediated attenuation of Tot levels on lymph gland growth in mxc mutants.
In addition, the authors tried to assess the intensity of effector caspase activation within LGs with non-specific antibodies.
In general, the manuscript has a potential to be improved, regardless of several flaws in experimental design identified.
Major problems
[1]
Strongly bimodal distributions presented on the Fig 7F for “positive area” data in all mxc experimental groups evidenced of several sources for signal that bias reported results severely.
Bergmann and others already reported that anti-caspase-3 antibody #9661 might detect unknown proteins unrelated to apoptotic changes, biasing results. Other technical apoptosis-irrelevant reasons to get bimodal distributions with the antibody #9661 are also possible (improper thresholds for detection of signal, ineffective RNAi constructs etc).
As a result, it is too preliminary to suggest any proapoptotic actions for Tot proteins on tumour cells based only on these data, without additional confirmatory evidences.
For more details on specificity of #9661 antibody, please see
Fan, Y., Bergmann, A. Cell Death Differ 17, 534–539 (2010). https://doi.org/10.1038/cdd.2009.185
Fogarty CE, Bergmann A. Methods Mol Biol. 2014;1133:109-117. doi:10.1007/978-1-4939-0357-3_7.
[1a]
The section “3.8. Fat Body-specific Depletion of Tot Genes Enhanced Apoptosis” must be omitted or revised extensively, since no strict evidences in favour of increased apoptosis were provided.
[1b]
If possible, additional unequivocal markers of apoptosis must be studied to validate the flawed conclusion.
[1c]
The abstract and the discussion sections must be changed with respect to non-specificity of antibodies applied.
[2]
It looks like that sequences applied for fat body-specific silencing of Tot transcripts (Sections 3.5, 3.8, 3.9) are not specific for targets assessed.
In particular, mentioned TotB-targeting construct is similar by sequence to some regions of TotA gene, while the TotF-targeting construct is similar to regions of Victoria(TotE) gene.
The specificity of applied RNAi constructs was not validated by PCR.
[2a]
Please indicate directly in the relevant Result sections 3.5, 3.8, 3.9 and on the Fig.5, 7, 8 and in the discussion that data reported might represent non-specific repression of several Tot genes at once.
[2b]
Please also discuss the possible consequences of such non-specificity of constructs applied within the Discussion section.
[3]
There are discrepancies between constructs reported in the section “3.5. Silencing TotB, TotC, or TotF gene by Expressing dsRNAs”, the Fig.5 and the section “2.1. Drosophila Stocks”
Please clarify.
[4]
The statistical reporting must be improved extensively.
[4a]
Where relevant, provide exact values for both significant and non-significant P values.
For ANOVAs, please provide F values and degrees of freedom.
For example, one-way ANOVA results should be reported like F(1, 510) = 6.71, p = 0.009
For t-tests, please provide t-values and degrees of freedom.
For example, Welch's t test should be reported as follows t(33) = 3.87, p = 0.007
For other examples, please see https://www.bachelorprint.eu/apa-style/reporting-statistics-in-apa/.
[4b]
Skewed data should be analysed by ANOVA or Welsch t-test only after applying relevant data transformations (power or Box-Cox).
[4c]
SD intervals should be reported instead of SEMs.
In case of skewed data, transformed SD borders should be reported.
[4d]
The data reported on the Fig. 4 should be analysed by the two-way ANOVA with factors of “genotype” and “dsRNA” assessed individually.
[5]
Please estimate and report the Chi-squares (or results of the Fischer Exact test) and CI95 confidence intervals for all data, which were considered as proportions.
Chi-squares must be reported within the text.
CI95 intervals better to illustrate as figures.
To achieve this, please use
the JASP statistical freeware (https://jasp-stats.org/, Frequencies module) ,
OpenEpi calculator (https://www.openepi.com/Menu/OE_Menu.htm),
the “Chi-squared test for r x c contingency table” tool (available at https://epitools.ausvet.com.au/chisq),
or another statistical service estimating probabilities and confidence intervals for RxC contingency tables.
[5a]
The reported shares on hemocytes and lamellocytes (section 3.1) provide no clues whether of these cells impacted the proportion primary and significantly.
To confirm the shift in favour of lamellocytes, please report the CI95 intervals for proportions of cells between conditions tested (in the form of the figure) and the Chi-square test results (within the text).
These data might help to illustrate that the increase in lamellocytes numbers might be the primary source of biased proportions.
[5b]
Similar, please estimate and report the Chi-squares and CI95 intervals for cell proportions reported in the sections 3.2, 3.6 and 3.7
[6]
Please do not mix up antimicrobial peptides and Turandot proteins. Better to use “proteins” for Turandot gene products.
Please improve the Abstract and the text respectively.
Minor issues
[7]
Please report (in the Supplement) the data on the decreasing viability in mutant larvae harbouring fat body-specific silencing of TotC.
[8]
The Fig. 2e panel is not illustrative for coepression of Hml and upd3 in mutant flies.
Please rearrange the Fig 2e elements using a bar chart design with two aligned different y-axes: one (for Hml) starting from the bottom-based zero and the second (for upd3) starting from the top-based zero.
Such a design will help to illustrate the suggestion of authors, which is not based on co-detection data currently.
[9]
The increase in the numbers of hemocytes demonstrating Stat92E-dependent fluorescence should be illustrated in a form of a bar chart as a separate panel for the Fig. 2.
[10]
The figure 3 and 4 panels are not illustrative due to the linear scale of y-axis on those panels.
Please improve the depiction of these panels by log10-transforming y-axes.
[11]
The statements done at [P6-LL236-238] are precocious when get ahead of the subsequent results. Please consider to improve or omit.
[12]
Language issues
[P10-LL326-327] “…decreased significantly by 3.9% of the TotA level, …1.4% for TotF, <0.2 % for TotB, and 1.2% for TotC” - Please avoid of ambiguous statements and improve.
[P15-L441] “…more hemocytes were localized on the mutant tumor LG.” – This statement is only suggestive. Please improve.
[P10-L324] “…quantitative real-time PCR… to quantify…” – Please improve
While AMP is a common abbreviation for antimicrobial peptides in the papers concerning with Drosophila immunity, it is easy to confuse AMP with for Adenosine monophosphate. Better to use AmPs, aMPs, AMPeps or smth like this for “antimicrobial peptides” entities.
[P16-LL480-481] “We scored phospho-H3 (pH3)-positive cells and converted them to 0.2 cells on average “ –Cells were not CONVERTED to 0.2 cells. The cell quantities were normalised. Please improve.
Author Response
Reviewer 1
Majo r issues [1]
Strongly bimodal distributions presented on the Fig 7F for “positive area” data in all mxc experimental
groups evidenced of several sources for signal that bias reported results severely.
Fan, Y., Bergmann, A. Cell Death Differ 17, 534-539 (2010)
https://doi.org/10.1038/cdd.2009.185
Fogarty CE, Bergmann A. Methods Mol Biol. 2014;1133:109-117. doi:10.1007/978-1-4939-
0357-3_7.
[1a]
The section “3.8. Fat Body-specific Depletion of Tot Genes Enhanced Apoptosis” must be omitted or
revised extensively, since no strict evidences in favour of increased apoptosis were provided.
(Autho rs’ respo nse )
We realized that the catalog number of the antibody used to detect the cells undergoing
apoptosis was wrong. We used the anti-cDcp1 antibody that can specifically recognize the
activated form of Dcp1, which is a Drosophila orthologue of the effector caspases (catalog
number # 9578, cell signaling technology). Therefore, we revise the sentence as follows:
“After several washes, LGs were immunostained with an anti-cleaved Dcp-1 antibody (1:200;
#9578, Cell Signaling Technology).”(line 183, page 5). We appreciate this reviewer for
his/her information that we need a special caution when using #9661 to detect the apoptosis
cells by immunostaining.
[1b]
If possible, additional unequivocal markers of apoptosis must be studied to validate the flawed
conclusion.
(Autho rs’ respo nse )
The anti-cDcp1 antibody (#9578, but not #9661) has been used as the specific antibody that
can recognize
Drosophila cells undergoing apoptosis, and several hundred papers in which
the antibody was used have been published so far (for example, Biomolecules 2022, 12(8),
1105; https://doi.org/10.3390/biom12081105). Therefore, we believe that our
immunostaining experiments using the anti-cDcp1 antibody can specifically highlight the
cells undergoing apoptosis. We corrected the catalog number of the antibody (line 183, page
5).
2
[1c ]
The abstract and the discussion sections must be changed with respect to non-specificity of antibodies
applied.
(Autho rs’ respo nse )
As we performed the immunostaining experiments shown in Fig. 7 using the anti -cDcp1
antibody (#9578) harboring a higher degree of specificity that can recognize the apoptotic
cells but did not use the #9661 antibody, we hope that it is not necessary to modify our
conclusion with respect to the effect of Tot proteins that induce apoptosis.
[2]
It looks like that sequences applied for fat body-specific silencing of Tot transcripts (Sections 3.5, 3.8,
3.9) are not specific for targets assessed.
In particular, mentioned TotB-targeting construct is similar by sequence to some regions of TotA gene,
while the TotF-targeting construct is similar to regions of Victoria(TotE) gene.
The specificity of applied RNAi constructs was not validated by PCR.
[2a]
Please indicate directly in the relevant Result sections 3.5, 3.8, 3.9 and on the Fig.5, 7, 8 and in the
discussion that data reported might represent non-specific repression of several Tot genes at once.
(Autho rs’ respo nse )
For the depletion of three Tot genes, TotA, TotB, and TotF, we used UAS-TotRNAi stocks which can
induce the expression of short dsRNAs against the relevant mRNA, respectively. When we compare
whole DNA sequences between each member in the Tot gene family, we found some sequences
showing similarity to other members, as the reviewer pointed. However, when we checked the
specificity score, s19 to evaluate the specificity of dsRNA sequences induced by each UAS-RNAi
stock, we have not found off-target sequences for TotB-targeting RNA sequences in other genes. The
RNA sequences with S19 values greater than 0.8 can be considered to have no off-targets( Dietzl et
al., 2007). Based on such criteria, we selected the UAS-RNAi stocks that can induce the dsRNA
sequences, which are regarded to have no off-targets, (https://shop.vbc.ac.at/vdrc_store/vdrc-fly-
stocks/rnai-libraries.html and https://www.flyrnai.org/up-torr/). By using the dsCheck
(https://dscheck.rnai.jp) and the BLAST search, we also confirmed that other mRNAs
showing a considerably high similarity in sequences (Expect Value
E<0.0001) with the
dsRNA sequences used for the
Tot gene silencing experiments are not transcribed in the larval
stage. As pointed by this reviewer, the TotF-targeting construct, P{GD3660}v8602 (VDRC #v8602)
indeed contains the nucleotide sequences similar to mRNA sequences of other Tot family gene,
Victoria (TotE) gene. However, the
TotE gene is barely expressed during the 3rd instar larval
stage. Thus, it is unlikely that dsRNA against TotF contributed even in part to the
3
enhancement of the LG hyperplasia in mxc mutant larvae (Fig. 5h,5i) via down-regulation of
TotE mRNA. We did not use
P{GD3660}v8602 but used
P{KK11099} (v107119) for
experiments in Fig. 7 and 8. Therefore, it is less likely to consider non-specific down-regulation of
multiple Tot genes simultaneously, although we cannot exclude the possibility. For further
confirmation of our conclusion, it may be effective to examine the effects of the knockdown of one
Tot gene on other Tot genes.
[2b]
Please also discuss the possible consequences of such non-specificity of constructs applied within the
Discussion section.
(Autho rs’ respo nse )
Based on the reasons described above, we believe it is less likely that dsRNA expression of one Tot
gene resulted in non-specific down-regulation of other Tot genes at once. Thus, we consider that the
enhancement of LG hyperplasia in mxcmbn1 larvae harboring the depletion of TotA, TotB or TotF results from
the specific down-regulation effect of each gene. Although at least one off-target, TotE is predicted in the
dsRNA sequence produced by TotFRNAi2, this gene is not expressed in the larval stage. Therefore, the
inhibitory effect of TotF depletion on the LG hyperplasia in mxc mutant is considered to be a consequence
of the downregulation of TotF. Similarly, suppression of apoptosis in the mutant LGs in Fig. 7, and
enhancement of mitotic cells in the mutant LGs in Fig. 8 are also responsible for the down-regulation of
these two Tot genes. However, we cannot exclude the possibility that simultaneous non-specific
downregulation of several Tot genes would influence the alteration of the phenotypes, as the reviewer
pointed. To strengthen our conclusion, it would be effective to investigate whether whole cDNA
sequences in which sequences used to produce the dsRNA are not included rescue the depletion effect
of other Tot genes in the mxc mutant larvae. We will include and clarify this issue in our next study.
As requested by the reviewer, we added the following sentences at the end of 4.4: “To investigate the
depletion effect of each Tot gene, we used the UAS-RNAi stocks that did not induce expression of
off-target genes [58]. However, we cannot completely exclude the possibility that simultaneous non-
specific downregulation of other Tot genes or non-related genes would be involved in the effect. To
strengthen our model, it would be important to investigate whether ectopic expression of whole cDNA,
in which the sequences used for the depletion are not contained, cancels the depletion effect of the Tot
genes” (line 688-694, page 22).
[3]
There are discrepancies between constructs reported in the section “3.5. Silencing TotB, TotC, or TotF
gene by Expressing dsRNAs”, the Fig.5 and the section “2.1. Drosophila Stocks”
Please clarify.
4
(Autho rs’ respo nse )
We apologize for the type error in the subtitle of 3.5. We performed the Tot gene silencing
experiments about TotA, B, and F in Fig. 5. We revised the sub-title 3.5 as follows: “3.5.
Silencing TotA, TotB, or TotF gene by Expressing dsRNAs Against the Relevant mRNAs in the Fat
Body Enhanced LG Hyperplasia in mxcmbn1 Larvae” (line 372, page 11). And the information
regarding the UAS-TotARNAi and UAS-TotCRNAi stocks was missing in the previous
manuscript. We added the following phrase, the sentence, and the references in which
previous studies showed the efficient depletion effects of the RNAi stocks in 2.1 Drosophila
stocks (line 95, page 3).
“For dsRNA-dependent gene silencing, the following UAS-RNAi stocks were used;
P{GD6210}v14416 [28] and P{KK112386}v106548 for TotA silencing (Vienna Drosophila Resource
Center (VDRC), Vienna, Austria), P{GD3091}v51123 and P{GD3091}v51124 for TotB silencing
(VDRC) [29], P{KK110624}v14420 for TotC silencing [30], P{GD3660}v8602 and for TotF silencing
(VDRC #v8602), and P{KK101199} (v107119)[31].” We appreciate this reviewer’s careful reading
of our manuscript and his/her valuable information.
[4] The statistical reporting must be improved extensively.
[4a]
Where relevant, provide exact values for both significant and non-significant P values. For ANOVAs,
please provide F values and degrees of freedom. For example, one-way ANOVA results should be
reported like F(1, 510) = 6.71, p = 0.009. For t-tests, please provide t-values and degrees of freedom.
https://www.bachelorprint.eu/apa-style/reporting-statistics-in-apa/.
(Autho rs’ respo nse )
As requested by reviewer 1, we first improved statistical methods extensively. We revised the
sentences in section 2.7. Statistical Analysis (lines 189-197, page 5). “Each dataset was assessed
using Welch’s
t-test , analysis of variance(ANOVA) or Fisher’s exact test. Data were tested
for normality by using Shapiro-Wilk test and normalized by Box-Cox common transforming
method. When the Box-Cox transformation could not be applied, we used the Yeo-Johnson
transformation. Subsequently, Welch ’ s
t-test or ANOVA was performed using the
transformed values. We used Welch’s
t-test for comparing the two groups. One-way ANOVA
followed by Bonferroni's multiple comparisons test was applied to analyze the differences in
more than two groups. Two-way ANOVA followed by Tukey’s multiple comparisons test was
performed to compare the mean differences between groups split into two independent
variables.”
5
We calculated exact p values for both significant and non-significant cases and showed each value
in Fig. 3a-d, Fig. 4a-d, Fig. 5i, Fig. 7f, Fig. 8f, Fig. S3a,b, Fig. S5 and the appropriate sections.
The results of one-way ANOVA in Fig. 5i can be described as F(7,190)=102.5, p = 4.66E-61, those in
Fig. 7f are described as F(4,95)=31.26, p=1.32 X 10-16, those in Fig.8 are F(4,99)=21.27, p=1.11 X
10-12. In Fig. 5i, the LG size differences of each genotype were compared using one-way ANOVA
followed by Bonferroni's multiple comparisons test and displayed. In Fig. 7f, the fluorescence
differences of LGs immunostained with anti-cDcp1 antibody between each genotype were compared
using one-way ANOVA followed by Bonferroni's multiple comparisons test and displayed. In
Fig. 8f, the fluorescence differences of LGs immunostained with anti-phospho-H3 antibody between
each genotype were compared using one-way ANOVA followed by Bonferroni's multiple
comparisons test and displayed.
For two-way ANOVA, we added the statistical results in Fig. 4 as follows:
TotA: effect of dsRNA; F (1, 8) = 248.9, p =2.60 X 10-7, effect of genotypes; F (1, 8) = 236.0, p =3.20
X 10-7, and effect of their interaction; F (1, 8) = 213.6, p =4.71 X 10-7.
TotB: effect of dsRNA; F (1, 8) = 478650, p = 2.13 X 10-20, effect of genotypes; F (1, 8) = 478626,
p = 2.13 X 10-20, and effect of their interaction; F (1, 8) = 476669, p=2.17 X 10-20.
TotC: effect of dsRNA; F (1, 8) = 4.836, p =0.059, effect of genotypes; F (1, 8) = 20.69, p =0.002,
and effect of their interaction; F (1, 8) = 5.296, p =0.050.
TotF: effect of dsRNA; F (1, 8) = 367.4, p = 5.69 X 10-08, and effect of genotypes; F (1, 8) = 323.0,
p =9.42 X 10-08, effect of their interaction; F (1, 8) = 315.0, p =1.04 X 10-07.
For Welch t-tests, we described t-values in Fig. 3a-d, Fig. S3 and Fig. S5 as follows:
As described in Fig.3, TotA: t(2.183)=19.20,p=0.0018, TotB: t(2.000)=9.698,p=0.0105,
TotC: t(2.000)=3.301,p=0.0938 and TotF: t(2.001)=5.144,p=0.0357.
When the significance probability value was less than the significance level value (0.05), each
conclusion and interpretation described in the text was considered significant.
However, for the qRT-PCR data in Fig. S3, the p-values for the experiments were beyond the
significant level we have set (0.05). t(2.099) = 1.524, p = 0.261 for TotB in Fig. S3 (line 707, page
22), t(2.054) = 2.235, p =0.1516 for TotF in Fig. S3. These data are also consistent with our
conclusion that non-specific dsRNA against GFP mRNA does not change the mRNA levels of TotB
or TotF.
As shown in Fig. S5C, to compare the percentage of the hemocytes containing TotF in control
and the
mxcmbn1 mutant larvae, we examined the statistical significance using Welch’s
t-test.
Consequently, the t-value was computed as follows; t(7.519) = 3.48,
p = 0.0091 .
[4b]
Skewed data should be analysed by ANOVA or Welsch t-test only after applying relevant data
6
transformations (power or Box-Cox).
(Autho rs’ respo nse )
In according with this comment of reviewer 1, we initially checked the data of each experiment to see
if they were normally distributed by the Shapiro-Wilk test. For data in Fig. 4c, Fig. 5i, Fig. 7f, and
Fig. 8f, the Box-Cox transformations were performed using a built-in function in the
scipy.stats module (ver. 1.11.1) of the Python (ver. 3.8.2) (https://www.python.org/downloads/). To
analyze the data in Fig. S5, the Box-Cox transformation could not be applied. Instead, we used the
Yeo-Jonson transformation. Subsequently, ANOVA or Welch’s t-tests were performed using the
transformed values. One-way ANOVA followed by Bonferroni's multiple comparisons test was
applied to analyze the differences in more than two groups. Two -way ANOVA followed by
Tukey’s multiple comparisons test was performed to compare the mean differences between
groups split into two independent variables. We described the methods in section 2.7.
[4c]
SD intervals should be reported instead of SEMs.
In case of skewed data, transformed SD borders should be reported.
(Autho rs’ respo nse )
According to the reviewer’s comment, we replaced the display of SEM with the SD intervals
in Fig. 3a-d, Fig. 4a-d, Fig. 5i, Fig. 7f, Fig. 8f, Fig.S3 and S5.
[4d]
The data reported on the Fig. 4 should be analysed by the two-way ANOVA with factors of “genotype”
and “dsRNA” assessed individually.
(Autho rs’ respo nse )
As requested by the reviewer, we revised the statistical calculation using two-way ANOVA in
the Fig. 4. In addition, we added sentences that explain the results analyzed by the two-way ANOVA
in the text (line 337-351, page 10) and the legend for Fig. 4 (line 369, page 11).
For the text, “The results of two-way analysis of variance (ANOVA) suggest that there is a
significant relationship between
TotA mRNA level and
Stat92E depletion (F(1,8) = 248.9,
p
= 2.60 X 10-7), between
TotA mRNA level and the genotype (
w and
mxcmbn1 )(F(1,8) = 236.0,
p =3.20 X 10-7), and their interaction (F(1,8) = 213.6,
p = 4.71 X 10-7. Consistently, there is
a significant relationship between
TotB mRNA level and
Stat92E depletion (F(1,8) = 478,650,
p = 2.13 X 10-20), and between
TotB mRNA level and the genotype (
w and
mxcmbn1 )(F(1,8)
= 4478,626,
p = 2.13 X 10-20), and their interaction (F(1,8) = 4476,669,
p =2.17 X 10-20.
There is also a significant relationship between
TotF mRNA level and
Stat92E depletion
7
(F(1,8) = 367.4,
p = 5.69 X 10-8), between
TotF mRNA level and the genotype (
w and
mxcmbn1)(F(1,8) = 323.0,
p = 9.42 X 10-8), and their interaction (F(1,8) =315.0,
p = 1.04 X
10-7). In contrast, there is no significant relationship between
TotC mRNA level and
Stat92E
depletion (F(1,8) = 44.836,
p = 0.06), neither the interaction between
Stat92E depletion and
the genotype (
w and
mxcmbn1)(F(1,8) = 5.296,
p = 0.05). However, there is a statistical
significance between
TotC mRNA level and the genotype (
w and
mxcmbn1) (F(1,8) = 220.69,
p = 0.002).”
[5]
Please estimate and report the Chi-squares (or results of the Fischer Exact test) and CI95 confidence
intervals for all data, which were considered as proportions.
Chi-squares must be reported within the text. CI95 intervals better to illustrate as figures.
[5a]
The reported shares on hemocytes and lamellocytes (section 3.1) provide no clues whether of these
cells impacted the proportion primary and significantly.
To confirm the shift in favour of lamellocytes, please report the CI95 intervals for proportions of cells
between conditions tested (in the form of the figure) and the Chi-square test results (within the text).
These data might help to illustrate that the increase in lamellocytes numbers might be the primary
source of biased proportions.
(Autho rs’ respo nse )
According to the reviewer’s suggestion for the data in section 3.1, we calculated the CI95
intervals for proportions of cells stained or unstained with the P1 or L1 antibody between w(control)
and mxc mutant larvae in Fig. S1e,f. According to the reviewer’s suggestion, we showed the
CI95 intervals for proportions of cells expressing the
hml or
upd3 between
w (control) and
mxc mutant larvae in Figure 1e,f. We performed Fisher’s exact test rather than Chi-square test. The
test using the proportion data estimated the p-value: p=3.11X10-5. Enough lower value than the
significant level value suggests that the hemocytes expressing P1/NimC1 and L1/Attila antigens,
which are plasmatocyte’s and lamellocyte’s markers, respectively, significantly increased in mxc
mutant larva, compared to normal control hemocytes. This is the evidence that many abnormal
hemocytes expressing the lamellocyte’s marker are differentiated in the hemolymph of the
mutant larvae without microbial infection. We added the following sentences in section 3.1 (line
218-220, page 6). “The statistical analysis suggests that the differences of genotype between
w
and
mxcmbn1 influenced the proportion of these two types of hemocytes in larval hemolymph
8
(Fisher’s exact test;
p = 3.11 X 10-5).“
[5b]
Similar, please estimate and report the Chi-squares and CI95 intervals for cell proportions reported in
the sections 3.2, 3.6 and 3.7
(Authors’ response)
According to the reviewer’s suggestion, we also performed Fisher’s exact test for proportions
of cells expressing the
hml or
upd3 between
w (control) and
mxc mutant larvae. Consequently,
the
p-value was estimated as p=4.44 X 10-16. This value suggests that the hemocytes
expressing the marker genes for mature hemocytes and those expressing an immature gene
significantly changed in mxc mutant larva, compared to normal control hemocytes. This
evidence supports our interpretation that many abnormal hemocytes expressing the immature
cell marker are contained in the hemolymph of the mutant larvae without microbial infection.
We added the following sentences in Results. “These data are consistent with our current
conclusion that some mutant hemocytes exhibit transformed phenotypes including defects in
cell differentiation” (line 245-246, page 6). For quantitative data on the hemocytes harboring
the STAT92E-GFP reporter in section 3.2, we have not performed detailed quantitative
analysis, but only a quick observation. We removed the percentage from the text (line 274 in
previous manuscript). We apologize for the mistake.
Regarding data in section 3.6, we performed Fisher’s exact test for proportions of circulating
hemocytes containing the TotF between w(control) and mxc mutant larvae , as requested by
reviewer 1. The p-value was estimated as p= 2.28 X 10-27 (for TotB), and p = 9.44 X 10-37 (for
TotF). The enough lower p-value estimated from the test supports our conclusion that much
more circulating hemocytes which contained the Tot B and TotF proteins were contained in
mxcmbn1 larvae than those in control hemocytes.
We added the following sentence at the end of section 3.6.: “The enough lower
p-values
support our conclusion that much more circulating hemocytes in mxcmbn1 larvae, which
contained the Tot B and F produced in the fat body than in control hemocytes.” (line 434-
439, page 14).
Regarding data in section 3.7 (Fig. S5C), we did not examine proportions of circulating
hemocytes containing the TotF on the LGs between w(control) and mxc mutant larvae.
Therefore, we performed the Welch’s
t-test, instead of Chi-square test as requested by
reviewer 1. The statement that we performed Student’s t-test in the previous manuscript was
our mistake. We corrected it. The enough lower p-value (p <0.01) than the significant value
9
supports our conclusion that significantly larger numbers of circulating hemocytes containing
the TotF protein adhered to the LGs in mxcmbn1 larvae than those in control larvae. According
to the reviewer’s suggestion, we added SD intervals, instead of SEM, in Figure S5C.
[6]
Please do not mix up antimicrobial peptides and Turandot proteins. Better to use “proteins” for
Turandot gene products. Please improve the Abstract and the text respectively.
(Autho rs’ respo nse )
As directed by the reviewer, we replaced all of the “Turandot peptides” that appeared at 25
locations in total, including the title with “Turandot proteins”.
Minor issues
[7] Please report (in the Supplement) the data on the decreasing viability in mutant larvae harbouring
fat body-specific silencing of TotC.
(Autho r’s respo nse )
Although the mRNA level of
TotC increased in the mxc mutant at the highest level among
the four Tot genes that express in larvae, the mxc mutant larvae harboring fat body-specific
depletion of TotC were obtained less than one-fourth of the mutant larvae having depletion
of TotB or TotF, possibly due to a strong enhancement of LG tumor or lower viability of the
relevant larvae for other reasons. We usually picked up 20-30 mxc mutant larvae harboring
the fat body-specific depletion of the Tot genes. It is not difficult for us to obtain enough
numbers of the mutant larvae with the depletion of TotA, B, or F for the immunostaining
experiments. In contrast, we only found five or fewer larvae harboring the TotC depletion
among a same-size population of larvae derived from the genetic crosses between r4-Gal4 and
the UAS-TotCRNAi in the mxc mutant background. Therefore, we selected TotB or TotF for
the experiments to investigate whether the fat body-specific depletion of the Tot mRNAs
influenced the LG tumor growth in the mxc mutant larvae, rather than TotC. As we have not
counted the total numbers of the target and sibling larvae in this experiment, unfortunately,
we cannot mention more details about the viability of the larvae with the corresponding
genotypes.
[8]
The Fig. 2e panel is not illustrative for coexpression of Hml and upd3 in mutant flies.
Please rearrange the Fig 2e elements using a bar chart design with two aligned different y-axes: one
(for Hml) starting from the bottom-based zero and the second (for upd3) starting from the top-based
10
zero. Such a design will help to illustrate the suggestion of authors, which is not based on co-detection
data currently.
(Autho rs’ respo nse )
As requested by the reviewer, we revised Fig. 1e (we think we should revise Fig. 1e, not Fig.
2e). We hope that the new figure will help the readers to understand our suggestion, which is not
currently based on the direct evidence indicating the co-expression of the Hml and Upd3.
[9]
The increase in the numbers of hemocytes demonstrating Stat92E-dependent fluorescence should be
illustrated in a form of a bar chart as a separate panel for the Fig. 2.
(Autho rs’ respo nse )
We performed quick observations to estimate which tissues or cells the Stat92E was activated
using the Stat92E-GFP reporter at the initial stage of this study. Therefore, we only checked
whether the GFP+ cells were present in circulating hemocytes from normal (no positive cells
were observed among more than 1500 cells from 20 larvae) or mutant larvae (approximately
90 % of the mutant cells (more than 900 cells from 20 mutant larvae were examined)). As we
have not obtained more quantitative data, we could not repeat the experiments in this short
period (10 days) for the revision.
[10]
The figure 3 and 4 panels are not illustrative due to the linear scale of y-axis on those panels.
(Autho rs’ respo nse )
As requested, we revised the y-axis of the graph to a logarithmic scale.
[11]
The statements done at [P6-LL236-238] are precocious when get ahead of the subsequent results.
Please consider to improve or omit.
(Autho rs’ respo nse )
We agreed that the statements described at line 236-238, page 6 in the previous manuscript are
precocious when get ahead of the subsequent results mentioned in 3.3. We removed the sentence.
[12]
[P10-LL326-327] “...decreased significantly by 3.9% of the TotA level, ...1.4% for TotF, <0.2 %
for TotB, and 1.2% for TotC” - Please avoid of ambiguous statements and improve.
(Autho rs’ respo nse )
Reviewer 2 Report
1- More detailed Pharmacological assays are necessary to support the hypothesis, however it may be explored as a future strategy.
2- Do the authors intend to examine other molecular routes for the tested peptides?
Author Response
1- More detailed Pharmacological assays are necessary to support the hypothesis, however it
may be explored as a f uture strategy.
(Authors’ response)
We agree with the review 2 that some pharmacological assays are also valid to further clarify
our hypothesis that macrophage-like hemocytes first recognize the tumor and then express
the inflammatory cytokine, Upd3. The Upd3-expressing hemocytes may migrate to the fat
body while secreting Upd3 in
Drosophila. We need to obtain a direct observation that
supports this point using live analysis in vitro or ex vivo assay. We further propose that t he
binding of Upd3 to its receptor, Dome on the fat body activates the JAK/STAT signaling
pathway and induces the transcription of
Tot genes. For this point, we would perform further
experiments to prove the involvement of JAK/STAT using some inhibitors of the pathway
such as methotrexate or aminopterin in vitro culture system or ex vivo system. We also have
experiment plan to clarify the important point of our hypothesis, how the released Tot
proteins induces apoptosis and suppress the proliferation of LG tumor cells. Tot polypeptides
are small proteins consisting of only 120-130 amino acids. We are now trying to synthesize
whole or partial polypeptides and to raise specific antibo dies. We hope that some
pharmacological assays using those experimental tools, if they work well, would be effective
to clarify the important issues in our hypothesis.
2- Do the authors intend to examine other molecular routes for the tested
peptides?
(Author’s response)
We have recently published another paper, in which we presented data that JNK, a stress-
activated kinase is highly activated in the lymph gland tumor in
mxcmbn1 mutant larvae
(Kinoshita et al., 2022, Biol Open. 2 0 2 2 11, bio059523. doi: 10.1242/bio.059523.).
Furthermore, we are now examining whether the JNK-mediated pathway is also activated in
circulating hemocytes in
mxcmbn1 mutant larvae. If this is the case, as it is known that the
activated JNK can induce expression of unpaired family proteins, it would possible to expect
that they are secreted from cells and activate the JAK/STAT pathway via binding of the
secreted unpaired proteins to the Domeless receptor on the surface of the fat body cells.
Once the receptor is activated, the Tot proteins can be synthesized and secreted from fat
body. It is reasonable to consider that the stress-activated JNK is activated in the mutant
circulating hemocytes. At the moment, we do not know whether the JNK activation is a nature of tumor cells or a normal response of the circulating hemocytes that recognized the tumor cells. To clarify this issue and understand the mechanism under the induction of Tot
expression in the hematopoietic tumor model in Drosophila, we have an experimental plant
to investigate whether the JNK pathway is activated in normal hemocytes that has recognized the LG tumor in vitro cultured system
Round 2
Reviewer 1 Report
Most critical issues have been resolved by the authors.
Nevertheless, several minor improvements are still required.
[1]
For the Fig7f and 8f, please identify the bimodal nature of distributions (identified for some of experimental groups) in the relevant subsections of the “Results” section.
In the “Discussion” section, please provide suggestions on the origins of bimodal distributions observed for indicators studied.
[2]
For TotC 2-way ANOVA results [LL448-450], the sentence “there is no significant relationship…neither the interaction between…” is misguiding. Please improve the text, indicating properly the tendencies identified.
[3]
Please report the exact p-values for “ns” results at the Fig S3.
[4]
Please improve the FigS1 F panel to delete technical elements print-screened.
[5]
Minor typos should be corrected across the text.
[L117] – word order
[L557] “mxcmbn1” superscript
etc
Author Response
[1] For the Fig7f and 8f, please identify the bimodal nature of distributions (identified for some of experimental groups) in the relevant subsections of the “Results” section.
According to the reviewer’s suggestion, we added three sentences to describe the bimodal distribution of apoptosis cell proportion shown in Fig. 7f (line 507-513, page 16-17).
Also, we added two sentences describing the bimodal distribution of mitotic cells shown in Fig. 8f (line 549-553, page 18).
In the “Discussion” section, please provide suggestions on the origins of bimodal distributions observed for indicators studied.
In accordance with the reviewer’s suggestion, we created a new section 4.4 in Discussion and added the following sentences to describe our idea on the origin of bimodal distributions for apoptosis cells and mitotic cells in the section, as follows:
“4.4 Possible origin of bimodal distribution of apoptosis and mitotic cells in LGs of mxcmbn1 and the mutant larvae harboring fat body-specific Tot inductionA clearly bimodal distribution of the apoptosis area was observed in the LGs of mxcmbn1 and the mutant larvae harboring fat body-specific depletion of TotB or TotF. We also noticed a bimodal distribution of mitotic cells in LGs of the mutant larvae harboring fat body-specific TotBRNAi, although the bimodality was exhibited less apparently. Even in the LGs of individual mutant larvae at almost the same development stage, the size varies widely among each individual larvae [7, 21]. Not all LG cells were transformed in the mutant larvae, and the frequencies of the tumor cells vary among larvae, consistent with the common nature known as intra-tumor heterogeneity [7, 22, 54]. The bimodal distributions allow us to speculate a possible threshold in the induction or action of Turandot. When a LG size exceeds a certain size limit, this may trigger the induction of the canonical innate immune pathways and JAK/STAT pathway in the fat body. When gram-negative bacteria are infected in Drosophila adults deficient for Toll or phagocytosis gene, a similar bimodal distribution of infection outcomes persists. The observed divergent infection is considered to be a natural result of mutual negative feedback between pathogens and the host immune response [55]. Additionally, substantial variation in mRNA levels of the AMP genes that are targets of the Toll- and/or Imd-mediated pathways is observed after bacterial infection or in mxcmbn1 larvae [7,23,55). Theoretical analyses suggested that small differences among hosts can be magnified into life-or-death differences to create a bimodality in infection outcomes [56]. Alternatively, as we discussed above, if Tot B and F succeed in arresting the cell cycle of LG tumor cells, some of the cells that have been inhibited from proliferation may be escaped from apoptosis. The two subpopulations of the tumor cells, which harbors a stronger and weaker immunostaining signal for activated effector caspase may correspond to the LG cells undergoing apoptosis and those assured survival, respectively. The LG cells exhibiting a stronger apoptosis signal might have been closely associated with the circulating hemocytes containing TotB or TotF and received the anti-tumor proteins. Further experiments need to be performed to clarify these interpretations. “(line 678-704, page 22).
[2] For TotC 2-way ANOVA results [LL448-450], the sentence “there is no significant relationship…neither the interaction between…” is misguiding. Please improve the text, indicating properly the tendencies identified.
As suggested by the reviewer, we revised the three sentences, as follows:
With regard to the TotC mRNA level, there is a statistical significance between the mRNA level and the genotype (w and mxcmbn1) (F(1,8) = 220.69, p = 0.002). In contrast, a significant relationship between TotC mRNA level and the Stat92E depletion was not observed (F(1,8) = 44.836, p = 0.059), although there is a tendency between them possibly due to a larger variation in the mRNA level.
There is also an interaction between Stat92E depletion and the genotype (w and mxcmbn1)(F(1,8) = 5.296, p = 0.050).” (line 346-351, page 10)
[3] Please report the exact p-values for “ns” results at the Fig S3. As requested, we described the exact p-values for ns results in Fig. S3.
[4] Please improve the FigS1 F panel to delete technical elements print-screened. As requested, we revised the inappropriate element in Fig. S1f panel.
[5] Minor typos should be corrected across the text. [L117] – word order [L557] “mxcmbn1” superscript
etc
As requested, we revised those typos in line 117-118, and in line 445.
